# PALU: KV-CACHE COMPRESSION WITH LOW-RANK PROJECTION

**Chi-Chih Chang**[1,3*]    **Wei-Cheng Lin**[1*]    **Chien-Yu Lin**[2*]
**Chong-Yan Chen**[1]   **Yu-Fang Hu**[1]   **Pei-Shuo Wang**[1]   **Ning-Chi Huang**[1]
**Luis Ceze**[2]   **Mohamed S. Abdelfattah**[3]   **Kai-Chiang Wu**[1]
[1]National Yang Ming Chiao Tung University   [2]University of Washington   [3]Cornell University

## ABSTRACT

Post-training KV-Cache compression methods typically either sample a subset of effectual tokens or quantize the data into lower numerical bit width. However, these methods cannot exploit redundancy in the hidden dimension of the KV tensors. This paper presents a hidden dimension compression approach called *Palu*, a KV-Cache compression framework that utilizes low-rank projection to reduce inference-time LLM memory usage. Palu decomposes the linear layers into low-rank matrices, caches compressed intermediate states, and reconstructs the full keys and values on the fly. To improve accuracy, compression rate, and efficiency, *Palu* further encompasses (1) a medium-grained low-rank decomposition scheme, (2) an efficient rank search algorithm, (3) low-rank-aware quantization compatibility enhancements, and (4) optimized GPU kernels with operators fusion. Extensive experiments with popular LLMs show that *Palu* compresses KV-Cache by 50%, while maintaining strong accuracy and delivering up to **1.89× speedup** on the RoPE-based attention module. When combined with quantization, *Palu*'s inherent quantization-friendly design yields small to negligible extra accuracy degradation, while saving additional memory than quantization-only methods and achieving up to **2.91× speedup** for the RoPE-based attention. Moreover, it maintains comparable or even **better accuracy (up to 1.19 lower perplexity)** compared to quantization-only methods. These results demonstrate *Palu*'s superior capability to effectively address the efficiency and memory challenges of LLM inference posed by KV-Cache. Our code is publicly available at: https://github.com/shadowpa0327/Palu.

## 1   INTRODUCTION

Large language models (LLMs) have propelled AI into new applications and capabilities, providing a high-level intelligence that previous machine learning (ML) models could not achieve. To speed up inference, caching the Key-Value states (KV-Cache) in memory is a simple yet effective technique. However, the size of the KV-Cache can grow rapidly, straining memory capacity and bandwidth especially with long context lengths (Fu, 2024); further, the memory-bounded nature of the decoding stage limits inference speed when loading KV-Cache data (Gholami et al., 2024). Therefore, KV-Cache compression has become a central research topic for running LLMs efficiently.

Although emerging attention mechanisms such as Multi-Query Attention (MQA) (Shazeer, 2019), Group-Query Attention (GQA) (Ainslie et al., 2023) and Multi-head Latent Attention (MLA) (DeepSeek-AI et al., 2024) can reduce KV-Cache size, it either requires model pre-training or has a significant impact on model's accuracy when converting from traditional Multi-Head Attention (MHA) (Chen et al., 2024). In contrast, post-training KV-Cache compression techniques offer an alternative approach to advance efficiency for existing models. Among various KV-Cache compression methods, quantization (Liu et al., 2024b; Hooper et al., 2024) and token eviction (Zhang et al., 2024; Xiao et al., 2024) stand out as effective strategies to reduce the memory footprint of KV-Cache.

Quantization methods aim to reduce the bit-width used to represent each piece of data, while token eviction techniques focus on retaining a partial set of KV-Cache. However, both methods neglect the

---

*Equal contribution

hidden dimensions of the KV-Cache, where substantial redundancy often resides. To capitalize on this untapped potential, we introduce *Palu*, a post-training KV-Cache compression framework that leverages low-rank projection to reduce the hidden dimension of KV tensors, offering an additional and orthogonal compression dimension to existing quantization and token eviction methods.

A naive way to utilize low-rank projection for compressing the KV-Cache is by directly mapping cached matrices into low-rank space (Jolliffe & Cadima, 2016; Zhao et al., 2024). However, this approach imposes an unacceptably heavy overhead of computing the decomposition matrices during runtime. To avoid this, *Palu statically decomposes the Key and Value-projection weight matrices* and *caches the latent representations* of the low-rank decomposition (see Fig. 1). This innovative design enables *Palu* to reduce memory while mitigating the runtime overhead of KV-Cache low-rank decomposition.

In designing an effective decomposition strategy for attention modules with multiple attention heads, we observed a clear trade-off between accuracy and reconstruction overhead. Decomposing the projection matrices across all attention heads together improves accuracy by preserving global information, but this approach significantly increases reconstruction costs. On the other hand, decomposing each head separately reduces reconstruction overhead but leads to a higher loss in accuracy. To address this, *Palu* introduces a medium-grained, group-head low-rank decomposition that strikes a balance between accuracy and reconstruction efficiency.

For LLMs, each linear projection module has a different sensitivity to compression (Sharma et al., 2023; Yuan et al., 2023). To exploit the sensitivity and improve accuracy, we design an efficient

Figure 1: *Palu*'s *low-rank projection method* for KV-Cache reduction. A weight matrix **W** of linear projection is decomposed into two low-rank matrices. Input **X** is down-projected to latent representation **H**, which is cached. **Y** can be reconstructed from **H** using the up-projection matrix **B**.

*rank search algorithm* based on Fisher information (Ly et al., 2017; Liu et al., 2021). Our algorithm automatically assigns a higher rank for important matrices and lower ranks for less critical ones, boosting accuracy at the same overall KV-Cache compression rate.

In addition to its low-rank decomposition, *Palu* is compatible with quantization techniques. We found that low-rank decomposition can introduce severe outliers in the latent representation, which significantly hinders accurate low-bit quantization. Although the Hadamard transformation has been shown to be effective for outlier elimination in recent studies (Tseng et al., 2024; Ashkboos et al., 2024b; Liu et al., 2024a; Chiang et al., 2024), its integration often introduces computational overhead during runtime. However, *Palu*'s inherent matrix pair structure makes it highly compatible with this technique, allowing the transformation matrices to be seamlessly fused into the forward and backward matrices, effectively mitigating outliers without impacting runtime efficiency.

We evaluate *Palu* on widely used LLMs and benchmarks. Our experiments demonstrate that *Palu* maintains strong zero-shot accuracy and perplexity with up to 50% low-rank compression. Moreover, when combining low-rank compression with quantization, *Palu* achieves an impressive **over 91.25% compression (11.4× reduction)** and yields a *significantly lower perplexity of 1.19* than KVQuant (Hooper et al., 2024), a state-of-the-art KV-Cache quantization method, which only achieves an 87.5% compression rate.

For latency evaluation, under a 50% KV-Cache compression rate without quantization, *Palu* demonstrates up to *1.89× and 2.2× speedup* for RoPE-based and non-RoPE attention modules. When integrated with quantization, *Palu* achieves up to *2.91× and 6.17× acceleration* on RoPE-based and non-RoPE attention, respectively. These results underscore *Palu*'s ability to significantly reduce KV-Cache memory footprint while boosting inference efficiency for LLMs.

Our key contributions include:

- *Palu*, a new post-training KV-Cache compression framework that caches *low-rank latent representations* of Key and Value states.

- *Group-head low-rank decomposition (G-LRD)*, an optimization for balancing accuracy and reconstruction efficiency.
- An *automated rank search algorithm* for adaptively assigning ranks to each decomposed matrix, given a target compression rate.
- A co-designed *quantization compatibility optimization* that eliminates low-rank-induced outliers and imposes zero runtime overhead.

## 2  BACKGROUND

### 2.1  MULTI-HEAD ATTENTION MECHANISM

The multi-head attention (MHA) mechanism (Vaswani et al., 2017) is a core component of the transformer architecture. Given a new input token $\mathbf{x} \in \mathbb{R}^d$, an MHA with $n$ heads projects the input into multiple queries, keys, and values using weight matrices $\mathbf{W}_i^q$, $\mathbf{W}_i^k$, and $\mathbf{W}_i^v$, respectively, for each head $i$, as shown by

$$\mathbf{q}_i = \mathbf{x}\mathbf{W}_i^q, \ \ \mathbf{k}_i = \mathbf{x}\mathbf{W}_i^k, \ \ \mathbf{v}_i = \mathbf{x}\mathbf{W}_i^v. \tag{1}$$

Here, $\mathbf{k}_i$ and $\mathbf{v}_i$ represent the key and value at time step $t$ for head $i$. We can then compute the attention score for each head $i$ and the corresponding attention output as

$$\mathbf{p}_{t,i} = \text{Softmax}\left(\frac{\mathbf{q}_i\mathbf{K}_i^T}{\sqrt{d_h}}\right), \ \ \mathbf{a}_i = \mathbf{p}_i\mathbf{V}_i, \tag{2}$$

where $\mathbf{K}_i$ and $\mathbf{V}_i$ denote the concatenation of current and all previous keys and values corresponding to the $i$-th head. The final MHA output is obtained by concatenating the outputs of all heads and then applying the out-projection layer $\mathbf{W}_o$, as shown by

$$\text{MHA}(\mathbf{x}) = \sum_{i=1}^{h}\mathbf{a}_i\mathbf{W}_i^o = \sum_{i=1}^{h}(\mathbf{p}_i\mathbf{V}_i)\mathbf{W}_i^o, \tag{3}$$

where $\mathbf{W}_i^o \in \mathbb{R}^{d_h \times d}$ represents the submatrices of the out-projection matrix for each head $i$.

### 2.2  SINGULAR VALUE DECOMPOSITION (SVD)

SVD (Jolliffe & Cadima, 2016) is a commonly used technique for computing the low-rank approximation for a given matrix. SVD decomposes a given matrix $\mathbf{W} \in \mathbb{R}^{m \times n}$ into three matrices: $\mathbf{W} = \mathbf{U}\mathbf{\Sigma}\mathbf{V}^T$. Here, $\mathbf{U}$ and $\mathbf{V}$ are orthogonal matrices containing the left and right singular vectors, respectively. The matrix $\mathbf{\Sigma}$ is a diagonal matrix that consists of singular values. After decomposition, the low-rank approximation of $\mathbf{W}$ can be described as

$$\mathbf{W} \approx \mathbf{A}\mathbf{B}, \ \ \mathbf{A} = \mathbf{U}_r\sqrt{\mathbf{\Sigma}_r}, \ \ \mathbf{B} = \sqrt{\mathbf{\Sigma}_r}\mathbf{V}_r^T, \tag{4}$$

where $\mathbf{A} \in \mathbb{R}^{m \times r}, \mathbf{B} \in \mathbb{R}^{r \times n}, \mathbf{\Sigma}_r \in \mathbb{R}^{r \times r}$. $\mathbf{\Sigma}_r$ is a diagonal matrix containing the largest $r$ singular values, and $\mathbf{U}_r, \mathbf{V}_r^T$ are corresponding singular vectors truncated from $\mathbf{U}$ and $\mathbf{V}^T$. This truncation and subsequent matrix formation let us approximate matrix $\mathbf{W}$ with two low-rank matrices $\mathbf{A}$ and $\mathbf{B}$, thereby reducing the storage by $\frac{mr+rn}{mn}$.

## 3  THE PALU FRAMEWORK

### 3.1  COMPRESSING THE KV-CACHE VIA LOW-RANK PROJECTION

A naïve approach to compress the KV-Cache with low-rank projection is to apply SVD directly on the KV-Cache and store the top-$r$ singular vectors. However, this approach poses significant computational overhead during runtime that makes it impractical for deployments (see Appendix H).

To apply low-rank projection more efficiently than directly decomposing the KV-Cache during runtime, *Palu* uses SVD to decompose the Key and Value projection matrices. This approach is based on the observation that low-rank decomposition rewrites the linear projection layer from $\mathbf{y} = \mathbf{x}\mathbf{W}$ into $\mathbf{y} = \mathbf{x}\mathbf{A}\mathbf{B}$.

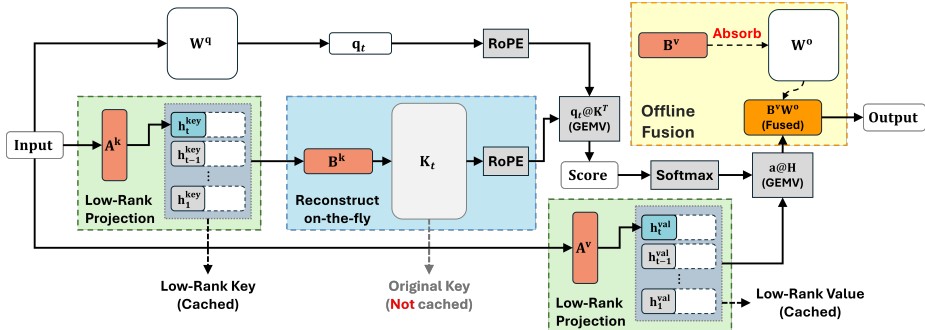

Figure 2: *Palu* uses low-rank decomposition ($\mathbf{W} \approx \mathbf{AB}$) to project the key (or value) to a lower-dimensional latent representation ($\mathbf{h}$), thereby reducing the size of the KV-Cache. The original key ($\mathbf{K}_t$) is reconstructed on-the-fly with $\mathbf{B}^k$, and $\mathbf{B}^v$ is fused into $\mathbf{W}^o$ to avoid reconstruction overhead. The fusion also reduces the computational burden for output projection.

Here, $\mathbf{A} \in \mathbb{R}^{d \times r}$ is the low-rank projection matrix, and $\mathbf{B} \in \mathbb{R}^{r \times d}$ is the reconstruction matrix derived by SVD. The forward process first *down-projects* the input token $\mathbf{x} \in \mathbb{R}^d$ into a low-dimensional latent space $\mathbf{h} \in \mathbb{R}^r$ and then *up-projects* it back to the original space:

$$\mathbf{h} = \mathbf{Ax}, \quad \mathbf{y} = \mathbf{Bh} \tag{5}$$

This two-step process lets *Palu* (1) store the lower dimension latent representation instead of the origin key and value states, and (2) reconstruct them during decoding.

### 3.1.1 INTEGRATION WITH THE ATTENTION MECHANISM AND OFFLINE MATRIX FUSION

We now describe how *Palu* decomposes the key and value linear layers for the attention mechanism. For each attention head $i$, *Palu* applies SVD and maps the key-projection matrix $\mathbf{W}_i^k$ and value-projection matrix $\mathbf{W}_i^v$ into $\mathbf{A}_i^k \mathbf{B}_i^k$ and $\mathbf{A}_i^v \mathbf{B}_i^v$.

Based on the formula of attention output in Eq. 2, *Palu* absorbs the reconstruction matrix $\mathbf{B}_i^v$ into the output projection matrix $\mathbf{W}_i^o$ offline:

$$\mathbf{a}_i \mathbf{W}_i^o = (\mathbf{p}_i \mathbf{V}_i) \mathbf{W}_i^o = (\mathbf{p}_i \mathbf{H}_i^v \mathbf{B}_i^v) \mathbf{W}_i^o = \mathbf{p}_i \mathbf{H}_i^v (\mathbf{B}_i^v \mathbf{W}_i^o) \tag{6}$$

Such fusion lets *Palu* skip the explicit reconstruction of the full value vectors, reduce the number of matrix multiplications, and improve efficiency. A similar approach applies for calculating attention scores. Matrix $\mathbf{B}_i^k$ can be fused into the query projection matrix $\mathbf{W}_i^q$ offline, as shown by

$$\mathbf{q}_i \mathbf{K}_i^T = \mathbf{q}_i (\mathbf{H}_i^k \mathbf{B}_i^k)^T = \mathbf{x}_t \mathbf{W}_i^q (\mathbf{B}_i^k)^T (\mathbf{H}_i^k)^T = \mathbf{x}_t \left( \mathbf{W}_i^q (\mathbf{B}_i^k)^T \right) (\mathbf{H}_i^k)^T. \tag{7}$$

Here, $\mathbf{B}_i^k \in \mathbb{R}^{r \times d_h}$ and $\mathbf{W}_i^q \in \mathbb{R}^{d \times d_h}$, so the fused matrix $(\mathbf{W}_i^q (\mathbf{B}_i^k)^T)$ has size $\mathbb{R}^{d \times r}$. This fusion boosts computational efficiency by reducing the matrix dimension during attention score calculation.

### 3.1.2 COMPABILITY WITH POSITIONAL EMBEDDING

Recent LLMs, such as the Llama family, apply Rotary Positional Embedding (*i.e.,* RoPE (Su et al., 2021)) onto the Query and Key states prior to their multiplication.

The non-linear nature of these positional embeddings prevents the matrix fusion of attention scores, as outlined in Eq. 7. To address this, *Palu* dynamically reconstructs the keys from latent representations on the fly. Specifically, *Palu* employs a custom GPU kernel that efficiently integrates key reconstruction, RoPE application, and subsequent Query-Key multiplication into a single fused operation. By transferring only the low-rank latent representations and performing reconstruction directly within GPU shared memory. By doing so, *Palu* substantially reduces the off-chip memory footprint, optimizing the memory-bound LLM decoding Yuan et al. (2024) process through a

memory-computation trade-off. Detailed implementation specifics of this kernel are provided in Appendix 4.1.

Note that for some positional embedding methods, such as ALiBi (Press et al., 2022), positional embedding is not directly applied to the Key states. Consequently, the fusion described in Eq. 7 remains valid. For these non-RoPE attention modules, *Palu* achieves greater speedup compared to RoPE-based attention, as their reconstruction can be avoided with matrix fusion.

## 3.2 DECOMPOSITION GRANULARITY

### 3.2.1 MULTI-HEAD LOW-RANK DECOMPOSITION

We name the per-head decomposition scheme in Sec. 3.1.1 as *multi-head low-rank decomposition (M-LRD)*. We found M-LRD often causes a non-negligible accuracy degradation (discussed further in Sec. 4.2), possibly because SVD fails to capture the common information shared across heads. Therefore, alternative approaches are needed to preserve model accuracy.

### 3.2.2 JOINT-HEAD LOW-RANK DECOMPOSITION

An alternative approach is to jointly decompose weight matrices for all heads. By considering the combined weight matrix $\mathbf{W}_{\text{joint}} = [\mathbf{W}_1, \mathbf{W}_2, \ldots, \mathbf{W}_n] \in \mathbb{R}^{d \times (d_h \cdot n_h)}$, we can perform a single low-rank decomposition $\mathbf{W}_{\text{joint}} \approx \mathbf{A}_{\text{joint}} \mathbf{B}_{\text{joint}}$, where $\mathbf{A}_{\text{joint}} \in \mathbb{R}^{d \times r_{\text{joint}}}$ and $\mathbf{B}_{\text{joint}} \in \mathbb{R}^{r_{\text{joint}} \times (d_h \cdot n_h)}$. We call this scheme *joint-head low-rank decomposition (J-LRD)*.

J-LRD has the advantage of preserving the common principal components shared among different heads. This occurs because SVD is particularly effective at capturing the dominant components when applied to a larger, combined matrix, resulting in a more accurate approximation.

For J-LRD, the joint latent representation shared among all heads can be computed with $\mathbf{h}_{\text{joint}} = \mathbf{x}\mathbf{A}_{\text{joint}}$. During decoding, the original states for each head can be reconstructed via

$$[\mathbf{y}_1, \ldots, \mathbf{y}_n] = \mathbf{h}_{\text{joint}} \mathbf{B}_{\text{joint}}.$$

Despite better-preserving model accuracy, J-LRD introduces *significant computational and memory overhead* during decoding. Specifically, the total number of floating point operations (FLOPs) to reconstruct the Key or Value state of all heads now becomes $L \cdot r_{\text{joint}} \cdot d_h \cdot n$. Assuming the same size as the total low-rank latent representations (*i.e.,* $r_{\text{joint}} = \sum_{i=1}^{n} r_i$), the total reconstruction cost is $n$ times higher than M-LRD, whose total FLOPs is $L \cdot r_i \cdot d_h \cdot n$. When considering the matrix fusion in Sec. 3.1.1, the fused matrix of J-LRD has a size of $r_{\text{joint}} \cdot d \cdot n$, which is also $n$ times larger than M-LRD, leading to substantial higher memory consumption.

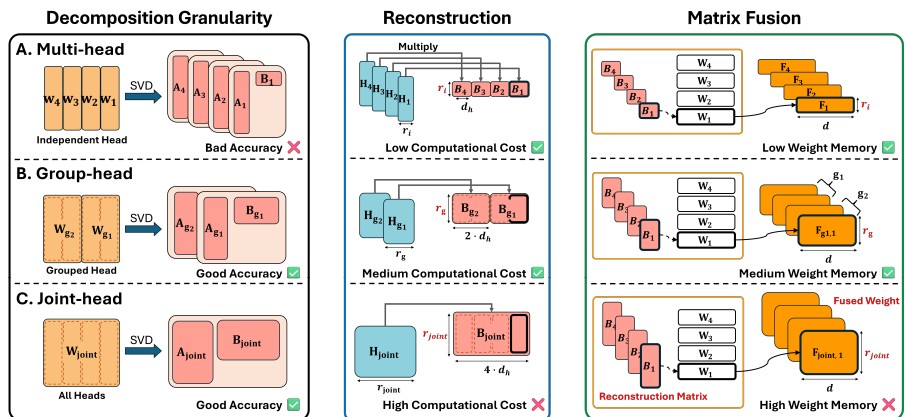

Figure 3: Performing decomposition at different granularities. Jointly decomposing multiple heads can achieve higher accuracy. Assuming the same total size of the latent representations (*i.e.,* $4 \cdot r_i = 2 \cdot r_g = r_{\text{joint}}$), the FLOPs for reconstruction overhead in joint-head decomposition schemes are 4 times larger than those in multi-head ones.

### 3.2.3 GROUP-HEAD LOW-RANK DECOMPOSITION

To balance the trade-off between accuracy and reconstruction cost, we propose *group-head low-rank decomposition* (G-LRD). G-LRD decomposes the matrices for a group of heads together. With combined weight matrices, it captures shared information within each group while limiting computational overhead and preserving accuracy.

To illustrate the G-LRD process, consider the weight matrices for a group of $s$ heads, $\mathbf{W}_{g_j} = [\mathbf{W}_{j,1} \dots \mathbf{W}_{j,s}]$, where $\mathbf{W}_{g_j} \in \mathbb{R}^{d \times (d_h \cdot s)}$. We low-rank decompose $\mathbf{W}_{g_j} \approx \mathbf{A}_{g_j} \mathbf{B}_{g_j}$, where $\mathbf{A}_{g_j} \in \mathbb{R}^{d \times r_g}$ and $\mathbf{B}_{g_j} \in \mathbb{R}^{r_g \times (d_h \cdot s)}$. The latent representation shared among attention heads in the same group can be computed as $\mathbf{h}_{\mathbf{g}_j} = \mathbf{x} \mathbf{U}_{\mathbf{g}_j}$. During decoding, the original key or value for each head can be reconstructed via

$$[\mathbf{y}_{j,1} \dots \mathbf{y}_{j,s}] = \mathbf{h}_{\mathbf{g}_j} \mathbf{B}_{\mathbf{g}_j}.$$

The FLOPs for reconstructing the keys and values for all heads in G-LRD is $L \cdot r_g \cdot d_h \cdot n$. Comparing the cost to J-LRD and assuming the same total rank size ($r_g \cdot n_g = r_{\text{joint}}$), G-LRD reduces the reconstruction cost by $n_g$. Similarly, G-LRD also reduces the fused matrix size by $n_g$. To sum up, G-LRD offers a middle ground between computation overhead and approximation accuracy. We illustrate M-LRD, J-LRD and G-LRD in Fig. 3. Please refer to Appendix C for further discussions on the costs of different decomposition granularities.

## 3.3 AUTOMATIC RANK ALLOCATION

To allocate an ideal rank size to the decomposition target, it is crucial to accurately estimate the importance of the target matrix (*e.g.,* grouped weights). In *Palu*, we identify **Fisher information** (Ly et al., 2017; Liu et al., 2021) as an accurate approximator since it can quantify the amount of information for each parameter. We then employ the sum of Fisher information to estimate the importance of the weight matrix of each linear layer (Abdelfattah et al., 2021).

Assuming that the compression sensitivity is proportional to Fisher information, we determine the rank for each weight matrix by computing the ratio of its Fisher information to the total Fisher information across all decomposition targets. We use this ratio to allocate the compression rate (*i.e.,* rank level $r$), ensuring that more important layers retain higher rank levels. For a detailed ablation study on our Automatic Rank Allocation, please refer to Appendix F.3.

## 3.4 QUANTIZATION COMPATIBILITY

We integrate quantization into *Palu* to compress the KV-Cache further. We observe that low-rank compressed latent representations have severe outliers, which limit quantization applicability in *Palu*. Unlike natural outliers described in previous KV-Cache quantization literature (Liu et al., 2024b; Hooper et al., 2024), these outliers are induced by SVD-based low-rank factorization.

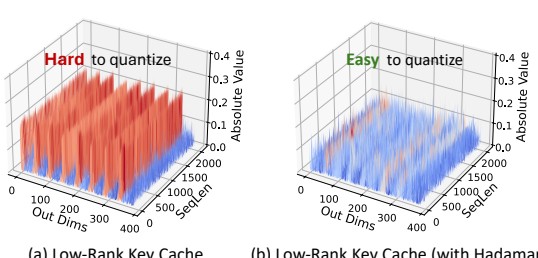

(a) Low-Rank Key Cache     (b) Low-Rank Key Cache (with Hadamard)

Figure 4: Activation distribution of the low-rank key caches at the $4^{th}$ Llama-2 attention layer.

Fig. 4 (a) shows the distribution of low-rank compressed key states from a layer of Llama-2 with G-LRD. Repeating outlier patterns appear at the beginning of each decomposed group because SVD arranges larger eigenvalues in the initial rows or columns, resulting in rapidly descending values in the latent representation. This pattern stretches the data distribution and hurts quantization accuracy.

Inspired by recent LLM quantization literature (Ashkboos et al., 2024b; Tseng et al., 2024), we apply the Walsh-Hadamard transform (WHT, Fino & Algazi) to eliminate outliers (Fig. 4 (b)), enabling a high quantization accuracy. However, this transformation introduces an extra matrix multiplication with associated runtime overhead. Unlike earlier methods (Ashkboos et al., 2024b) that must apply online WHT when quantizing KV-Cache, we optimize this process by integrating the Hadamard matrix into low-rank decomposed weights with no additional compute overhead, as described by

$$\mathbf{W} \approx \mathbf{AB} = (\mathbf{AR})(\mathbf{R}^T\mathbf{B}) = \hat{\mathbf{A}}\hat{\mathbf{B}}, \tag{8}$$

where $\mathbf{R}$ is the Hadamard matrix. This optimization allows *Palu* to integrate the proposed low-rank compression technique with low-bit quantization. Our experiments show that, on top of the low-rank compression, our quantization method only *negligibly increases perplexity*, even at extreme levels such as *3-bit or 2-bit* with a simple per-token quantization scheme (see Sec. 4.3).

# 4 EXPERIMENTS

## 4.1 EXPERIMENTS SETUP

**Models and Tasks.** We evaluate *Palu* on four LLM families, Llama-2 (Touvron et al., 2023), Llama-3 (Dubey et al., 2024), Mistral (Jiang et al., 2023) and LongChat (Li et al., 2023). For accuracy evaluation, we measure perplexity on the WikiText-2 (Merity et al., 2016) and C4 (Raffel et al., 2020) datasets and use LM-Evaluation-Harness (Gao et al., 2023) to measure zero-shot accuracy on six common sense tasks. We also evaluate long context accuracy on 16 tasks in LongBench (Bai et al., 2023). Unless specification, we refer to baseline as a model with non-compressed KV-Cache. See Appendix G for further details on the dataset and settings.

**Compression Settings.** We implemented *Palu* based on the Huggingface library (Wolf et al., 2020). Decomposition of the Key and Value projection layers was performed using the truncation-aware SVD method proposed by SVD-LLM (Wang et al., 2024). Unless otherwise specified, *Palu*'s results are G-LRD with a group size of 4 (gs-4), with equal rank size for each group. To calculate Fisher information in rank searching, we used 2048 random samples from Wikitext-2, each with a sequence length of 1024. For quantization integration in *Palu*, we use a simple per-token, asymmetric integer quantization. For evaluation on quantization results, we compare *Palu* to advanced KV-Cache quantization methods, including Atom (Zhao et al., 2023), KVQaunt (Hooper et al., 2024), and KIVI (Liu et al., 2024b). Refer to Sec. 5 for a brief summary of these methods.

**GPU Kernels Implementation.** We implemented a customized kernel for attention score with reconstruction in Triton (Tillet et al., 2019) (See Appendix B). For quantization integration, we implemented kernels in CUDA for attention output and non-RoPE attention score, where matrix fusion can be applied (refer to Sec. 3.1.1 and Fig. 2). Our low-precision kernel fuses the dequantization process and the follow-up multiplication with low-rank compressed keys or values, enabling efficient processing on quantized latent KV-Cache. When evaluating speedup with quantization, we compare to the non-compressed baseline and KIVI (Liu et al., 2024b), which we use their official code[*] in our experiments.

## 4.2 RESULTS WITH DIFFERENT DECOMPOSITION GRANULARITY

We evaluate perplexity and zero-shot accuracy of *Palu* with a **50% low-rank compression rate** using M-LRD, G-LRD, and J-LRD on Llama2-7B and Llama3-8B-Instruct, and present the results in Table 1.

Table 1: Perplexity and zero-shot accuracy of *Palu* at 50% compression rate.

| Model | Method | Perplexity ↓ | | Zero-Shot Accuracy (%) ↑ | | | | | | |
|---|---|---|---|---|---|---|---|---|---|---|
| | | Wiki2 | C4 | OBQA | Hella | PIQA | ARC-e | ARC-c | Wino | Avg. |
| Llama2-7B | Baseline | 5.47 | 7.26 | 44.20 | 76.00 | 78.07 | 76.30 | 46.42 | 69.30 | 65.05 |
| | J-LRD | 5.62 | 7.75 | 45.40 | 75.57 | 77.48 | 75.97 | 45.31 | 69.22 | 64.82 |
| | G-LRD | 6.01 | 9.82 | 43.60 | 73.39 | 76.33 | 73.02 | 42.57 | 66.77 | 62.61 |
| | M-LRD | 6.75 | 12.01 | 39.60 | 65.35 | 74.76 | 67.17 | 35.24 | 64.64 | 57.79 |
| Llama3-8B-Inst | Baseline | 8.28 | 13.01 | 43.20 | 75.80 | 78.62 | 81.61 | 56.83 | 71.90 | 67.99 |
| | J-LRD | 9.12 | 15.90 | 43.40 | 73.20 | 76.50 | 79.63 | 51.96 | 72.45 | 66.19 |
| | G-LRD | 10.11 | 17.87 | 42.60 | 70.36 | 76.06 | 76.30 | 48.99 | 72.38 | 64.45 |
| | M-LRD | 12.38 | 23.02 | 38.80 | 63.04 | 73.67 | 69.78 | 42.58 | 62.51 | 58.40 |

**Perplexity Evaluation.** As Table 1 shows, for the Llama2-7B model, *Palu*'s M-LRD method fails to maintain a low perplexity at a 50% compression rate. In contrast, despite having a high recomputation cost, J-LRD significantly outperforms M-LRD and achieves a 5.62 perplexity on WikiText-2.

---

[*]https://github.com/jy-yuan/KIVI

For G-LRD, which still maintains a low computation cost, yields a 6.01 perplexity on Wikitext-2, showing a great balance between model accuracy and compression overheads. The same trend is observed in the Llama-3-8B model as well. More results Llama-2-13B can be found in Appendix E.

**Zero-shot Evaluation Results.** Similar to the perplexity evaluation, the J-LRD method demonstrates the best performance for the zero-shot accuracy on Llama-2-7B, with only a 0.23% average accuracy degradation. M-LRD method results in the lowest average performance, with a 7.26% drop in accuracy compared to the baseline. In comparison, G-LRD only has a 2.4% average accuracy decline, offering a sweet spot between model accuracy and compression overheads again.

## 4.3 RESULTS OF QUANTIZATION INTEGRATION

Table 2 showcases the impact of quantization on perplexity and KV-Cache size when combined with *Palu*. With 3-bit quantization, *Palu* incurs only a **slight 0.08 and 0.23 perplexity increase** at 30% and 50% low-rank compression rate. These demonstrate a minimal accuracy trade-off for significant compression gains compared to the 16-bit baseline. Notably, at 2-bit quantization, *Palu* decisively outperforms the state-of-the-art KVQuant method, **reducing perplexity by 1.19 and 0.54**, while further slashing memory usage by 30% and 50%. These results establish *Palu* with quantization as a superior KV-Cache compression method.

Table 2: Quantization perplexity and KV-Cache size for Llama2-7B on WikiText-2. For perplexity, sequence length is 4096. KV-Cache size is demonstrated for 128K sequence length.

| Method | Bit | PPL | KV-Cache Size (GB) | Comp. Rate |
|---|---|---|---|---|
| Baseline | 16 | 5.12 | 64.0 | - |
| Palu-30% | 16 | 5.25 | 44.8 | 30% |
| Palu-50% | 16 | 5.63 | 32.0 | 50% |
| Atom | 3 | 6.15 | 12.6 | 80.32% |
| KVQuant | 3 | 5.35 | 12.0 | 81.25% |
| Palu-30% | 3 | **5.33** | 8.4 | **86.87%** |
| Palu-50% | 3 | 5.77 | 6.0 | 90.63% |
| Atom | 2 | 117.88 | 8.6 | 86.56% |
| KVQuant | 2 | 6.95 | 8.0 | 87.50% |
| Palu-30% | 2 | **5.76** | 5.6 | **91.25%** |
| Palu-50% | 2 | **6.41** | 4.0 | **93.75%** |

## 4.4 EVALUATION ON LONG CONTEXT DATASETS

Table 3: Experiment Results on LongBench: The average bit widths represent the total storage cost per element in the compressed KV-Cache, including the overhead of quantization parameters. These values are calculated for each approach, assuming a context length of 10K.

| Model | Method | Avg. Bits | Comp. Ratio | Multi-QA | Single-QA | Summa-rization | Few-Shot | Code | Synthetic | Avg. |
|---|---|---|---|---|---|---|---|---|---|---|
| | Baseline | 16 | 1.00x | 29.63 | 36.43 | 28.10 | 66.71 | 54.16 | 44.87 | 42.54 |
| Mistral-7B-v0.2 | Palu-30% | 16 | 1.43x | 29.83 | 36.52 | 27.48 | 65.70 | 55.16 | 37.92 | 41.55 |
| | Palu-50% | 16 | 2.00x | 26.92 | 35.33 | 26.01 | 64.04 | 44.54 | 16.88 | 36.23 |
| | KIVI-2 | 3.16 | 5.05x | 28.81 | 35.07 | 27.60 | 66.45 | 54.47 | 40.28 | 41.45 |
| | Palu-30% (3 bits) | 3.13 | 7.59x | 29.48 | 36.40 | 27.20 | 65.73 | 53.19 | 34.74 | 40.77 |
| | Palu-50% (3 bits) | 3.13 | 10.6x | 26.73 | 32.72 | 25.73 | 63.25 | 44.43 | 18.57 | 35.71 |
| | Baseline | 16 | 1.00x | 23.95 | 31.12 | 26.74 | 63.80 | 56.91 | 15.25 | 36.32 |
| LongChat-7B-v1.5 | Palu-30% | 16 | 1.43x | 22.42 | 29.43 | 25.52 | 62.87 | 58.99 | 14.25 | 35.45 |
| | Palu-50% | 16 | 2.00x | 22.61 | 25.33 | 25.52 | 60.12 | 43.52 | 6.84 | 30.82 |
| | KIVI-2 | 3.16 | 5.06x | 23.24 | 30.19 | 26.47 | 63.54 | 53.51 | 16.13 | 35.60 |
| | Palu-30% (3 bits) | 3.13 | 7.59x | 23.12 | 29.21 | 25.04 | 61.99 | 54.38 | 11.25 | 34.33 |
| | Palu-50% (3 bits) | 3.13 | 10.6x | 18.56 | 24.14 | 22.35 | 58.76 | 40.50 | 6.02 | 29.03 |

To access *Palu*'s ability for long-context scenarios, we evaluate baseline, KIVI and *Palu*'s accuracy on LongBench (Bai et al., 2023) Here, we evaluate the Mistral-7B and LongChat-7B models, which have up to 32K context length. We report the average score for each task type separately, as well as the overall average across all 16 tasks. The results are shown in Table 3. We report the accuracy of KIVI using the configuration with a group size of 32 and 128-element fp16 residual (Liu et al., 2024b).

As Table 3 indicates, we find that at a 50% low-rank compression level, *Palu* is relatively difficult to fully preserve accuracy. However, at a 30% compression level, *Palu* achieves only a minor average accuracy drop ($< 1\%$) compared to the baseline for both models. Furthermore, *Palu* can quantize the low-rank latent KV-Cache down to 3 bits, with less than 1% further accuracy degradation. Overall, *Palu* maintains a strong 40.77% and 34.33% average accuracy for Mistral-7B and LongChat-7B, with an impressive 7.59x compression ratio. Compared to KIVI (Liu et al., 2024b), *Palu* achieves a similar accuracy, while having an additional 30% compression rate from low-rank. Notably, *Palu*

does not require the complex grouped quantization and mixed-precision techniques employed by KIVI, resulting in a high inference efficiency (see Sec. 4.5 for details).

## 4.5 LATENCY EVALUATION

In this section, we provide latency and speedup evaluation, using Llama-2-7b as the base model. We measure decode latency on a single RTX 4090 GPU and compare *Palu* to the FP16 and KIVI-4-bit baselines. We evaluate *Palu*'s latency at a 50% compression rate, where we set compression rates for key and value to 75% and 25%, respectively. This allocation is based on our observations from the rank allocation results (see Appendix F.3 for details). For the FP16 baseline, we use the default implementation from HuggingFace. For KIVI, we use the CUDA kernels from its official repository. Due to the small memory capacity of RTX 4090 GPU, we adopt a 4-bit quantization (Frantar et al., 2024) for the weights of all linear layers. Our results are the average of 100 runs.

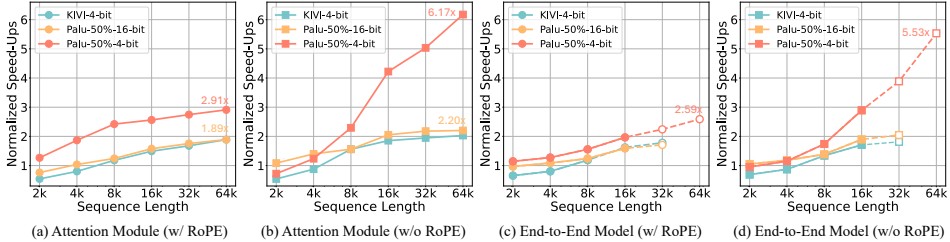

(a) Attention Module (w/ RoPE)  (b) Attention Module (w/o RoPE)  (c) End-to-End Model (w/ RoPE)  (d) End-to-End Model (w/o RoPE)

Figure 5: Normalized speedup for both the attention module and end-to-end model decoding. Solid lines represent exact measurements, while dashed lines indicate the FP16 baselines are out of memory, and the speedups are compared to the estimated baseline's latency.

### 4.5.1 SPEEDUPS OF ATTENTION MODULE AND END-TO-END DECODING.

**Attention module speedup.** We compare latency against standard attention without compression or quantization and show the speedups of *Palu* and KIVI-4-bit in Fig. 5 (a) and (b) for RoPE-based and non-RoPE attention. For RoPE-based attention, we applied our online reconstruction kernel for key and employed offline fusion for value as described in Sec 3.1.2.

As shown in Fig. 5 (a), *Palu* has minimal to no speedup when the sequence length is short, e.g. 4K. However, as sequence length increases, *Palu* delivers substantial performance gains. At 64K input length, *Palu* achieves a **1.89× speedup** over the FP16 baseline when using low-rank projection alone. By further applying 4-bit quantization to the Value states, the **speedup rises to 2.91×** for the same 64K context length, owing to our optimized low-precision kernel and reduced memory loading times. This performance notably surpasses KIVI-4-bit, which only achieves a 1.89× speedup at 64K, hindered by the overheads of its fined-grained group quantization. Notably, for RoPE-based attention, *Palu*-4-bit does not quantize key, as our online reconstruction kernel only supports FP16 precision for now.

For non-RoPE attention, we apply matrix fusion to both the Key and Value states (Eq. 7), effectively eliminating all reconstruction overhead. At a 64K sequence length with a 50% compression rate, *Palu* achieves a **2.20× speedup over the FP16 baseline**. By further applying 4-bit quantization to both the Key and Value states, *Palu* boosts the speedup to **6.17×** for 64K input length. These results demonstrate that combining low-rank compression and quantization significantly enhances inference efficiency, particularly in long-context scenarios.

**End-to-end speedup.** We present the end-to-end speedups in Fig. 5 (c) and (d), measuring the decoding latency of generating the next token at various input lengths. Similar to the attention performance results, *Palu* shows minimal or no speedup for short sequences but delivers significant acceleration for longer sequences. Without quantization, *Palu* achieves up to **1.71× and 2.05× speedups** for RoPE-based and non-RoPE models, respectively. With a 50% compression rate, *Palu* runs up to 32K input length on an RTX 4090 GPU. By incorporating 4-bit quantization, *Palu* handles even longer sequences and delivers **2.59× and 5.53× end-to-end speedups** at a 64K sequence length. *Palu* integrated with quantization provides a substantial speed advantage over KIVI-4-bit, which only reaches 1.78× and 1.81× speedups at 32K sequence length for RoPE and non-RoPE scenarios, respectively, and is out-of-memory for longer sequences.

### 4.5.2 KERNEL FOR ROPE-BASED ATTENTION SCORE

In this section, we evaluate the performance of our online reconstruction kernel for RoPE-based attention scores. We measure latency from the **pre-RoPE query vector** to **post-GEMV attention score**, and compare it with PyTorch's GEMV, which is used in the baseline attention (see Fig. 2).

We present speedups for group size 1, 4, and 32 at different sequence lengths in Fig. 6. For gs-32 (J-LRD), the highest accuracy decomposition, the high reconstruction cost causes a significant slowdown across all sequence lengths. For gs-1 (M-LRD), our kernel achieves up to a 3.56× speedup at sequence length 16K, showing strong performance when moderate accuracy loss is acceptable. For gs-4 (G-LRD), our kernel reaches up to 1.95× speedup. These results emphasize the need to explore various decomposition granularities for better accuracy and speed tradeoffs.

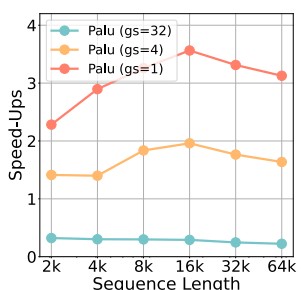

Figure 6: Speedup of *Palu*'s attention score kernel with online reconstruction.

We also observe that speedup decreases for sequence lengths beyond 16K due to rising reconstruction costs, shifting the online reconstruction from memory- to compute-bound. A potential optimization is to quantize the decomposed weight matrices further and leverage high-throughput, low-precision hardware (*e.g.,* INT4 Tensor Cores) for online reconstruction, which we leave for future work. Despite the speedup drop at longer lengths, *Palu*'s overall attention speedup increases with longer input, thanks to matrix fusion on the Value state and the reduced memory footprint.

## 5 RELATED WORK

**SVD for LLM Compression.** Several works have explored using SVD to compress LLMs. An early approach (Noach & Goldberg, 2020) applied standard SVD to weight matrices, resulting in significant compression errors. FWSVD (Hsu et al., 2022) addressed this by using Fisher information to prioritize parameters, while ASVD (Yuan et al., 2023) considered activation outliers. SVD-LLM (Wang et al., 2024) further minimized compression loss for each singular value. Unlike these methods, which compress model weights, *Palu* focuses on reducing KV-Cache size.

**KV-Cache Quantization.** Quantization is a widely used technique for compressing KV-Cache. Atom (Zhao et al., 2023) applies simple per-token quantization, while WKVQuant (Yue et al., 2024) introduces a two-level scheme to enhance accuracy. KIVI (Liu et al., 2024b) uses per-channel and per-token quantization for Keys and Values, combined with ultra fine-grained group quantization. KVQuant (Hooper et al., 2024) employs a similar setup but incorporates non-uniform quantization and sparse matrices to handle outliers. On top of these approaches, GEAR (Kang et al., 2024) adds a low-rank matrix to compensate for quantization errors. In *Palu*, we leverage low-rank techniques to exploit hidden dimension redundancy and achieve outstanding compression through simple per-token quantization.

**MLA.** The recently released DeepSeek-V2 model (DeepSeek-AI et al., 2024) introduces the MLA mechanism, which reduces KV-Cache size by down-projecting Key and Value to a low-rank space and reconstructing them to full rank at runtime. Although MLA may seem similar to *Palu* at a high level, particularly with J-LRD, our design and derivation processes are fundamentally different. Unlike MLA, a new attention mechanism requiring pre-training, *Palu* is specifically designed for post-training integration. *Palu* focuses on converting existing models with MHA or GQA to support low-rank compressed KV-Cache, preserving high accuracy while enhancing inference efficiency.

## 6 CONCLUSION

We introduce *Palu*, a novel KV-Cache compression framework that decomposes linear projection weight matrices and caches the compressed latent representations. We propose various optimizations, including group-head low-rank decomposition, automatic rank allocation algorithm, quantization compatibility enhancement, and customized kernels with operator fusion. With these optimizations, *Palu* can maintain accuracy while achieving significant memory reduction and high inference speedup.

A​CKNOWLEDGEMENTS

This research is supported in part by Taiwan's NSTC under Grant No. 113-2640-E-A49-004 and the National Science Foundation under Grant No. 2339084. Luis Ceze is supported by the Lazowska Endowed Professorship. We would like to express our appreciation to Sandy Kaplan from the University of Washington for her invaluable assistance in editing this paper.

R​EFERENCES

Mohamed S Abdelfattah, Abhinav Mehrotra, Łukasz Dudziak, and Nicholas Donald Lane. Zero-cost proxies for lightweight NAS. 2021. URL `https://openreview.net/forum?id=0cmMMy8J5q`.

Muhammad Adnan, Akhil Arunkumar, Gaurav Jain, Prashant Nair, Ilya Soloveychik, and Purushotham Kamath. Keyformer: Kv cache reduction through key tokens selection for efficient generative inference. *Proceedings of Machine Learning and Systems*, 7, 2024.

Joshua Ainslie, James Lee-Thorp, Michiel de Jong, Yury Zemlyanskiy, Federico Lebrón, and Sumit Sanghai. Gqa: Training generalized multi-query transformer models from multi-head checkpoints. *arXiv preprint arXiv:2305.13245*, 2023.

Saleh Ashkboos, Maximilian L. Croci, Marcelo Gennari do Nascimento, Torsten Hoefler, and James Hensman. SliceGPT: Compress large language models by deleting rows and columns. In *The Twelfth International Conference on Learning Representations*, 2024a. URL `https://openreview.net/forum?id=vXxardq6db`.

Saleh Ashkboos, Amirkeivan Mohtashami, Maximilian L Croci, Bo Li, Martin Jaggi, Dan Alistarh, Torsten Hoefler, and James Hensman. Quarot: Outlier-free 4-bit inference in rotated llms. *arXiv preprint arXiv:2404.00456*, 2024b.

Yushi Bai, Xin Lv, Jiajie Zhang, Hongchang Lyu, Jiankai Tang, Zhidian Huang, Zhengxiao Du, Xiao Liu, Aohan Zeng, Lei Hou, Yuxiao Dong, Jie Tang, and Juanzi Li. Longbench: A bilingual, multitask benchmark for long context understanding. *arXiv preprint arXiv:2308.14508*, 2023.

Yonatan Bisk, Rowan Zellers, Jianfeng Gao, Yejin Choi, et al. Piqa: Reasoning about physical commonsense in natural language. In *Proceedings of the AAAI conference on artificial intelligence*, volume 34, pp. 7432–7439, 2020.

Patrick Chen, Hsiang-Fu Yu, Inderjit Dhillon, and Cho-Jui Hsieh. Drone: Data-aware low-rank compression for large nlp models. In M. Ranzato, A. Beygelzimer, Y. Dauphin, P.S. Liang, and J. Wortman Vaughan (eds.), *Advances in Neural Information Processing Systems*, volume 34, pp. 29321–29334. Curran Associates, Inc., 2021. URL `https://proceedings.neurips.cc/paper_files/paper/2021/file/f56de5ef149cf0aedcc8f4797031e229-Paper.pdf`.

Yuang Chen, Cheng Zhang, Xitong Gao, Robert D. Mullins, George A. Constantinides, and Yiren Zhao. Optimised grouped-query attention mechanism for transformers, 2024. URL `https://arxiv.org/abs/2406.14963`.

Hung-Yueh Chiang, Chi-Chih Chang, Natalia Frumkin, Kai-Chiang Wu, and Diana Marculescu. Quamba: A post-training quantization recipe for selective state space models, 2024. URL `https://arxiv.org/abs/2410.13229`.

Peter Clark, Isaac Cowhey, Oren Etzioni, Tushar Khot, Ashish Sabharwal, Carissa Schoenick, and Oyvind Tafjord. Think you have solved question answering? try arc, the ai2 reasoning challenge. *arXiv:1803.05457v1*, 2018.

Pradeep Dasigi, Kyle Lo, Iz Beltagy, Arman Cohan, Noah A. Smith, and Matt Gardner. A dataset of information-seeking questions and answers anchored in research papers. In Kristina Toutanova, Anna Rumshisky, Luke Zettlemoyer, Dilek Hakkani-Tur, Iz Beltagy, Steven Bethard, Ryan Cotterell, Tanmoy Chakraborty, and Yichao Zhou (eds.), *Proceedings of the 2021 Conference of*

*the North American Chapter of the Association for Computational Linguistics: Human Language Technologies*, pp. 4599–4610, Online, June 2021. Association for Computational Linguistics. doi: 10.18653/v1/2021.naacl-main.365. URL https://aclanthology.org/2021.naacl-main.365.

DeepSeek-AI, Aixin Liu, Bei Feng, Bin Wang, Bingxuan Wang, Bo Liu, Chenggang Zhao, Chengqi Dengr, Chong Ruan, Damai Dai, Daya Guo, Dejian Yang, Deli Chen, Dongjie Ji, Erhang Li, Fangyun Lin, Fuli Luo, Guangbo Hao, Guanting Chen, Guowei Li, H. Zhang, Hanwei Xu, Hao Yang, Haowei Zhang, Honghui Ding, Huajian Xin, Huazuo Gao, Hui Li, Hui Qu, J. L. Cai, Jian Liang, Jianzhong Guo, Jiaqi Ni, Jiashi Li, Jin Chen, Jingyang Yuan, Junjie Qiu, Junxiao Song, Kai Dong, Kaige Gao, Kang Guan, Lean Wang, Lecong Zhang, Lei Xu, Leyi Xia, Liang Zhao, Liyue Zhang, Meng Li, Miaojun Wang, Mingchuan Zhang, Minghua Zhang, Minghui Tang, Mingming Li, Ning Tian, Panpan Huang, Peiyi Wang, Peng Zhang, Qihao Zhu, Qinyu Chen, Qiushi Du, R. J. Chen, R. L. Jin, Ruiqi Ge, Ruizhe Pan, Runxin Xu, Ruyi Chen, S. S. Li, Shanghao Lu, Shangyan Zhou, Shanhuang Chen, Shaoqing Wu, Shengfeng Ye, Shirong Ma, Shiyu Wang, Shuang Zhou, Shuiping Yu, Shunfeng Zhou, Size Zheng, T. Wang, Tian Pei, Tian Yuan, Tianyu Sun, W. L. Xiao, Wangding Zeng, Wei An, Wen Liu, Wenfeng Liang, Wenjun Gao, Wentao Zhang, X. Q. Li, Xiangyue Jin, Xianzu Wang, Xiao Bi, Xiaodong Liu, Xiaohan Wang, Xiaojin Shen, Xiaokang Chen, Xiaosha Chen, Xiaotao Nie, Xiaowen Sun, Xiaoxiang Wang, Xin Liu, Xin Xie, Xingkai Yu, Xinnan Song, Xinyi Zhou, Xinyu Yang, Xuan Lu, Xuecheng Su, Y. Wu, Y. K. Li, Y. X. Wei, Y. X. Zhu, Yanhong Xu, Yanping Huang, Yao Li, Yao Zhao, Yaofeng Sun, Yaohui Li, Yaohui Wang, Yi Zheng, Yichao Zhang, Yiliang Xiong, Yilong Zhao, Ying He, Ying Tang, Yishi Piao, Yixin Dong, Yixuan Tan, Yiyuan Liu, Yongji Wang, Yongqiang Guo, Yuchen Zhu, Yuduan Wang, Yuheng Zou, Yukun Zha, Yunxian Ma, Yuting Yan, Yuxiang You, Yuxuan Liu, Z. Z. Ren, Zehui Ren, Zhangli Sha, Zhe Fu, Zhen Huang, Zhen Zhang, Zhenda Xie, Zhewen Hao, Zhihong Shao, Zhiniu Wen, Zhipeng Xu, Zhongyu Zhang, Zhuoshu Li, Zihan Wang, Zihui Gu, Zilin Li, and Ziwei Xie. Deepseek-v2: A strong, economical, and efficient mixture-of-experts language model, 2024.

Abhimanyu Dubey, Abhinav Jauhri, Abhinav Pandey, Abhishek Kadian, Ahmad Al-Dahle, Aiesha Letman, Akhil Mathur, Alan Schelten, Amy Yang, Angela Fan, Anirudh Goyal, Anthony Hartshorn, Aobo Yang, Archi Mitra, Archie Sravankumar, Artem Korenev, Arthur Hinsvark, Arun Rao, Aston Zhang, Aurelien Rodriguez, Austen Gregerson, Ava Spataru, Baptiste Roziere, Bethany Biron, Binh Tang, Bobbie Chern, Charlotte Caucheteux, Chaya Nayak, Chloe Bi, Chris Marra, Chris McConnell, Christian Keller, Christophe Touret, Chunyang Wu, Corinne Wong, Cristian Canton Ferrer, Cyrus Nikolaidis, Damien Allonsius, Daniel Song, Danielle Pintz, Danny Livshits, David Esiobu, Dhruv Choudhary, Dhruv Mahajan, Diego Garcia-Olano, Diego Perino, Dieuwke Hupkes, Egor Lakomkin, Ehab AlBadawy, Elina Lobanova, Emily Dinan, Eric Michael Smith, Filip Radenovic, Frank Zhang, Gabriel Synnaeve, Gabrielle Lee, Georgia Lewis Anderson, Graeme Nail, Gregoire Mialon, Guan Pang, Guillem Cucurell, Hailey Nguyen, Hannah Korevaar, Hu Xu, Hugo Touvron, Iliyan Zarov, Imanol Arrieta Ibarra, Isabel Kloumann, Ishan Misra, Ivan Evtimov, Jade Copet, Jaewon Lee, Jan Geffert, Jana Vranes, Jason Park, Jay Mahadeokar, Jeet Shah, Jelmer van der Linde, Jennifer Billock, Jenny Hong, Jenya Lee, Jeremy Fu, Jianfeng Chi, Jianyu Huang, Jiawen Liu, Jie Wang, Jiecao Yu, Joanna Bitton, Joe Spisak, Jongsoo Park, Joseph Rocca, Joshua Johnstun, Joshua Saxe, Junteng Jia, Kalyan Vasuden Alwala, Kartikeya Upasani, Kate Plawiak, Ke Li, Kenneth Heafield, Kevin Stone, Khalid El-Arini, Krithika Iyer, Kshitiz Malik, Kuenley Chiu, Kunal Bhalla, Lauren Rantala-Yeary, Laurens van der Maaten, Lawrence Chen, Liang Tan, Liz Jenkins, Louis Martin, Lovish Madaan, Lubo Malo, Lukas Blecher, Lukas Landzaat, Luke de Oliveira, Madeline Muzzi, Mahesh Pasupuleti, Mannat Singh, Manohar Paluri, Marcin Kardas, Mathew Oldham, Mathieu Rita, Maya Pavlova, Melanie Kambadur, Mike Lewis, Min Si, Mitesh Kumar Singh, Mona Hassan, Naman Goyal, Narjes Torabi, Nikolay Bashlykov, Nikolay Bogoychev, Niladri Chatterji, Olivier Duchenne, Onur Çelebi, Patrick Alrassy, Pengchuan Zhang, Pengwei Li, Petar Vasic, Peter Weng, Prajjwal Bhargava, Pratik Dubal, Praveen Krishnan, Punit Singh Koura, Puxin Xu, Qing He, Qingxiao Dong, Ragavan Srinivasan, Raj Ganapathy, Ramon Calderer, Ricardo Silveira Cabral, Robert Stojnic, Roberta Raileanu, Rohit Girdhar, Rohit Patel, Romain Sauvestre, Ronnie Polidoro, Roshan Sumbaly, Ross Taylor, Ruan Silva, Rui Hou, Rui Wang, Saghar Hosseini, Sahana Chennabasappa, Sanjay Singh, Sean Bell, Seohyun Sonia Kim, Sergey Edunov, Shaoliang Nie, Sharan Narang, Sharath Raparthy, Sheng Shen, Shengye Wan, Shruti Bhosale, Shun Zhang, Simon Vandenhende, Soumya Batra, Spencer Whitman, Sten Sootla, Stephane Collot, Suchin Gururangan, Sydney

Borodinsky, Tamar Herman, Tara Fowler, Tarek Sheasha, Thomas Georgiou, Thomas Scialom, Tobias Speckbacher, Todor Mihaylov, Tong Xiao, Ujjwal Karn, Vedanuj Goswami, Vibhor Gupta, Vignesh Ramanathan, Viktor Kerkez, Vincent Gonguet, Virginie Do, Vish Vogeti, Vladan Petrovic, Weiwei Chu, Wenhan Xiong, Wenyin Fu, Whitney Meers, Xavier Martinet, Xiaodong Wang, Xiaoqing Ellen Tan, Xinfeng Xie, Xuchao Jia, Xuewei Wang, Yaelle Goldschlag, Yashesh Gaur, Yasmine Babaei, Yi Wen, Yiwen Song, Yuchen Zhang, Yue Li, Yuning Mao, Zacharie Delpierre Coudert, Zheng Yan, Zhengxing Chen, Zoe Papakipos, Aaditya Singh, Aaron Grattafiori, Abha Jain, Adam Kelsey, Adam Shajnfeld, Adithya Gangidi, Adolfo Victoria, Ahuva Goldstand, Ajay Menon, Ajay Sharma, Alex Boesenberg, Alex Vaughan, Alexei Baevski, Allie Feinstein, Amanda Kallet, Amit Sangani, Anam Yunus, Andrei Lupu, Andres Alvarado, Andrew Caples, Andrew Gu, Andrew Ho, Andrew Poulton, Andrew Ryan, Ankit Ramchandani, Annie Franco, Aparajita Saraf, Arkabandhu Chowdhury, Ashley Gabriel, Ashwin Bharambe, Assaf Eisenman, Azadeh Yazdan, Beau James, Ben Maurer, Benjamin Leonhardi, Bernie Huang, Beth Loyd, Beto De Paola, Bhargavi Paranjape, Bing Liu, Bo Wu, Boyu Ni, Braden Hancock, Bram Wasti, Brandon Spence, Brani Stojkovic, Brian Gamido, Britt Montalvo, Carl Parker, Carly Burton, Catalina Mejia, Changhan Wang, Changkyu Kim, Chao Zhou, Chester Hu, Ching-Hsiang Chu, Chris Cai, Chris Tindal, Christoph Feichtenhofer, Damon Civin, Dana Beaty, Daniel Kreymer, Daniel Li, Danny Wyatt, David Adkins, David Xu, Davide Testuggine, Delia David, Devi Parikh, Diana Liskovich, Didem Foss, Dingkang Wang, Duc Le, Dustin Holland, Edward Dowling, Eissa Jamil, Elaine Montgomery, Eleonora Presani, Emily Hahn, Emily Wood, Erik Brinkman, Esteban Arcaute, Evan Dunbar, Evan Smothers, Fei Sun, Felix Kreuk, Feng Tian, Firat Ozgenel, Francesco Caggioni, Francisco Guzmán, Frank Kanayet, Frank Seide, Gabriela Medina Florez, Gabriella Schwarz, Gada Badeer, Georgia Swee, Gil Halpern, Govind Thattai, Grant Herman, Grigory Sizov, Guangyi, Zhang, Guna Lakshminarayanan, Hamid Shojanazeri, Han Zou, Hannah Wang, Hanwen Zha, Haroun Habeeb, Harrison Rudolph, Helen Suk, Henry Aspegren, Hunter Goldman, Ibrahim Damlaj, Igor Molybog, Igor Tufanov, Irina-Elena Veliche, Itai Gat, Jake Weissman, James Geboski, James Kohli, Japhet Asher, Jean-Baptiste Gaya, Jeff Marcus, Jeff Tang, Jennifer Chan, Jenny Zhen, Jeremy Reizenstein, Jeremy Teboul, Jessica Zhong, Jian Jin, Jingyi Yang, Joe Cummings, Jon Carvill, Jon Shepard, Jonathan McPhie, Jonathan Torres, Josh Ginsburg, Junjie Wang, Kai Wu, Kam Hou U, Karan Saxena, Karthik Prasad, Kartikay Khandelwal, Katayoun Zand, Kathy Matosich, Kaushik Veeraraghavan, Kelly Michelena, Keqian Li, Kun Huang, Kunal Chawla, Kushal Lakhotia, Kyle Huang, Lailin Chen, Lakshya Garg, Lavender A, Leandro Silva, Lee Bell, Lei Zhang, Liangpeng Guo, Licheng Yu, Liron Moshkovich, Luca Wehrstedt, Madian Khabsa, Manav Avalani, Manish Bhatt, Maria Tsimpoukelli, Martynas Mankus, Matan Hasson, Matthew Lennie, Matthias Reso, Maxim Groshev, Maxim Naumov, Maya Lathi, Meghan Keneally, Michael L. Seltzer, Michal Valko, Michelle Restrepo, Mihir Patel, Mik Vyatskov, Mikayel Samvelyan, Mike Clark, Mike Macey, Mike Wang, Miquel Jubert Hermoso, Mo Metanat, Mohammad Rastegari, Munish Bansal, Nandhini Santhanam, Natascha Parks, Natasha White, Navyata Bawa, Nayan Singhal, Nick Egebo, Nicolas Usunier, Nikolay Pavlovich Laptev, Ning Dong, Ning Zhang, Norman Cheng, Oleg Chernoguz, Olivia Hart, Omkar Salpekar, Ozlem Kalinli, Parkin Kent, Parth Parekh, Paul Saab, Pavan Balaji, Pedro Rittner, Philip Bontrager, Pierre Roux, Piotr Dollar, Polina Zvyagina, Prashant Ratanchandani, Pritish Yuvraj, Qian Liang, Rachad Alao, Rachel Rodriguez, Rafi Ayub, Raghotham Murthy, Raghu Nayani, Rahul Mitra, Raymond Li, Rebekkah Hogan, Robin Battey, Rocky Wang, Rohan Maheswari, Russ Howes, Ruty Rinott, Sai Jayesh Bondu, Samyak Datta, Sara Chugh, Sara Hunt, Sargun Dhillon, Sasha Sidorov, Satadru Pan, Saurabh Verma, Seiji Yamamoto, Sharadh Ramaswamy, Shaun Lindsay, Shaun Lindsay, Sheng Feng, Shenghao Lin, Shengxin Cindy Zha, Shiva Shankar, Shuqiang Zhang, Shuqiang Zhang, Sinong Wang, Sneha Agarwal, Soji Sajuyigbe, Soumith Chintala, Stephanie Max, Stephen Chen, Steve Kehoe, Steve Satterfield, Sudarshan Govindaprasad, Sumit Gupta, Sungmin Cho, Sunny Virk, Suraj Subramanian, Sy Choudhury, Sydney Goldman, Tal Remez, Tamar Glaser, Tamara Best, Thilo Kohler, Thomas Robinson, Tianhe Li, Tianjun Zhang, Tim Matthews, Timothy Chou, Tzook Shaked, Varun Vontimitta, Victoria Ajayi, Victoria Montanez, Vijai Mohan, Vinay Satish Kumar, Vishal Mangla, Vítor Albiero, Vlad Ionescu, Vlad Poenaru, Vlad Tiberiu Mihailescu, Vladimir Ivanov, Wei Li, Wenchen Wang, Wenwen Jiang, Wes Bouaziz, Will Constable, Xiaocheng Tang, Xiaofang Wang, Xiaojian Wu, Xiaolan Wang, Xide Xia, Xilun Wu, Xinbo Gao, Yanjun Chen, Ye Hu, Ye Jia, Ye Qi, Yenda Li, Yilin Zhang, Ying Zhang, Yossi Adi, Youngjin Nam, Yu, Wang, Yuchen Hao, Yundi Qian, Yuzi He, Zach Rait, Zachary DeVito, Zef Rosnbrick, Zhaoduo Wen, Zhenyu Yang, and Zhiwei Zhao. The llama 3 herd of models, 2024. URL https://arxiv.org/abs/2407.21783.

Alexander Fabbri, Irene Li, Tianwei She, Suyi Li, and Dragomir Radev. Multi-news: A large-scale multi-document summarization dataset and abstractive hierarchical model. In Anna Korhonen, David Traum, and Lluís Màrquez (eds.), *Proceedings of the 57th Annual Meeting of the Association for Computational Linguistics*, pp. 1074–1084, Florence, Italy, July 2019. Association for Computational Linguistics. doi: 10.18653/v1/P19-1102. URL `https://aclanthology.org/P19-1102`.

Fino and Algazi. Unified matrix treatment of the fast walsh-hadamard transform. *IEEE Transactions on Computers*, C-25(11):1142–1146, 1976. doi: 10.1109/TC.1976.1674569.

Elias Frantar, Roberto L. Castro, Jiale Chen, Torsten Hoefler, and Dan Alistarh. Marlin: Mixed-precision auto-regressive parallel inference on large language models, 2024. URL `https://arxiv.org/abs/2408.11743`.

Yao Fu. Challenges in deploying long-context transformers: A theoretical peak performance analysis, 2024. URL `https://arxiv.org/abs/2405.08944`.

Leo Gao, Jonathan Tow, Baber Abbasi, Stella Biderman, Sid Black, Anthony DiPofi, Charles Foster, Laurence Golding, Jeffrey Hsu, Alain Le Noac'h, Haonan Li, Kyle McDonell, Niklas Muennighoff, Chris Ociepa, Jason Phang, Laria Reynolds, Hailey Schoelkopf, Aviya Skowron, Lintang Sutawika, Eric Tang, Anish Thite, Ben Wang, Kevin Wang, and Andy Zou. A framework for few-shot language model evaluation, 12 2023. URL `https://zenodo.org/records/10256836`.

Suyu Ge, Yunan Zhang, Liyuan Liu, Minjia Zhang, Jiawei Han, and Jianfeng Gao. Model tells you what to discard: Adaptive KV cache compression for LLMs. In *The Twelfth International Conference on Learning Representations*, 2024. URL `https://openreview.net/forum?id=uNrFpDPMyo`.

Amir Gholami, Zhewei Yao, Sehoon Kim, Coleman Hooper, Michael W. Mahoney, and Kurt Keutzer. Ai and memory wall, 2024. URL `https://arxiv.org/abs/2403.14123`.

Bogdan Gliwa, Iwona Mochol, Maciej Biesek, and Aleksander Wawer. Samsum corpus: A human-annotated dialogue dataset for abstractive summarization. *CoRR*, abs/1911.12237, 2019. URL `http://arxiv.org/abs/1911.12237`.

Daya Guo, Canwen Xu, Nan Duan, Jian Yin, and Julian J. McAuley. Longcoder: A long-range pre-trained language model for code completion. In Andreas Krause, Emma Brunskill, Kyunghyun Cho, Barbara Engelhardt, Sivan Sabato, and Jonathan Scarlett (eds.), *International Conference on Machine Learning, ICML 2023, 23-29 July 2023, Honolulu, Hawaii, USA*, volume 202 of *Proceedings of Machine Learning Research*, pp. 12098–12107. PMLR, 2023. URL `https://proceedings.mlr.press/v202/guo23j.html`.

Coleman Hooper, Sehoon Kim, Hiva Mohammadzadeh, Michael W Mahoney, Yakun Sophia Shao, Kurt Keutzer, and Amir Gholami. Kvquant: Towards 10 million context length llm inference with kv cache quantization. *arXiv preprint arXiv:2401.18079*, 2024.

Yen-Chang Hsu, Ting Hua, Sungen Chang, Qian Lou, Yilin Shen, and Hongxia Jin. Language model compression with weighted low-rank factorization. In *International Conference on Learning Representations*, 2022. URL `https://openreview.net/forum?id=uPv9Y3gmAI5`.

Edward J Hu, yelong shen, Phillip Wallis, Zeyuan Allen-Zhu, Yuanzhi Li, Shean Wang, Lu Wang, and Weizhu Chen. LoRA: Low-rank adaptation of large language models. In *International Conference on Learning Representations*, 2022. URL `https://openreview.net/forum?id=nZeVKeeFYf9`.

Albert Q. Jiang, Alexandre Sablayrolles, Arthur Mensch, Chris Bamford, Devendra Singh Chaplot, Diego de las Casas, Florian Bressand, Gianna Lengyel, Guillaume Lample, Lucile Saulnier, Lélio Renard Lavaud, Marie-Anne Lachaux, Pierre Stock, Teven Le Scao, Thibaut Lavril, Thomas Wang, Timothée Lacroix, and William El Sayed. Mistral 7b, 2023.

Ian T Jolliffe and Jorge Cadima. Principal component analysis: a review and recent developments. *Philosophical transactions of the royal society A: Mathematical, Physical and Engineering Sciences*, 374(2065):20150202, 2016.

Mandar Joshi, Eunsol Choi, Daniel Weld, and Luke Zettlemoyer. TriviaQA: A large scale distantly supervised challenge dataset for reading comprehension. In Regina Barzilay and Min-Yen Kan (eds.), *Proceedings of the 55th Annual Meeting of the Association for Computational Linguistics (Volume 1: Long Papers)*, pp. 1601–1611, Vancouver, Canada, July 2017. Association for Computational Linguistics. doi: 10.18653/v1/P17-1147. URL `https://aclanthology.org/P17-1147`.

Hao Kang, Qingru Zhang, Souvik Kundu, Geonhwa Jeong, Zaoxing Liu, Tushar Krishna, and Tuo Zhao. Gear: An efficient kv cache compression recipe for near-lossless generative inference of llm, 2024.

Dacheng Li, Rulin Shao, Anze Xie, Ying Sheng, Lianmin Zheng, Joseph Gonzalez, Ion Stoica, Xuezhe Ma, and Hao Zhang. How long can context length of open-source llms truly promise? In *NeurIPS 2023 Workshop on Instruction Tuning and Instruction Following*, 2023.

Xin Li and Dan Roth. Learning question classifiers. In *19th International Conference on Computational Linguistics, COLING 2002, Howard International House and Academia Sinica, Taipei, Taiwan, August 24 - September 1, 2002*, 2002. URL `https://aclanthology.org/C02-1150/`.

Yuhong Li, Yingbing Huang, Bowen Yang, Bharat Venkitesh, Acyr Locatelli, Hanchen Ye, Tianle Cai, Patrick Lewis, and Deming Chen. SnapKV: LLM knows what you are looking for before generation. In *The Thirty-eighth Annual Conference on Neural Information Processing Systems*, 2024. URL `https://openreview.net/forum?id=poE54GOq2l`.

Liyang Liu, Shilong Zhang, Zhanghui Kuang, Aojun Zhou, Jing-Hao Xue, Xinjiang Wang, Yimin Chen, Wenming Yang, Qingmin Liao, and Wayne Zhang. Group fisher pruning for practical network compression. In Marina Meila and Tong Zhang (eds.), *Proceedings of the 38th International Conference on Machine Learning, ICML 2021, 18-24 July 2021, Virtual Event*, volume 139 of *Proceedings of Machine Learning Research*, pp. 7021–7032. PMLR, 2021. URL `http://proceedings.mlr.press/v139/liu21ab.html`.

Tianyang Liu, Canwen Xu, and Julian McAuley. Repobench: Benchmarking repository-level code auto-completion systems, 2023.

Zechun Liu, Changsheng Zhao, Igor Fedorov, Bilge Soran, Dhruv Choudhary, Raghuraman Krishnamoorthi, Vikas Chandra, Yuandong Tian, and Tijmen Blankevoort. Spinquant–llm quantization with learned rotations. *arXiv preprint arXiv:2405.16406*, 2024a.

Zirui Liu, Jiayi Yuan, Hongye Jin, Shaochen Zhong, Zhaozhuo Xu, Vladimir Braverman, Beidi Chen, and Xia Hu. Kivi: A tuning-free asymmetric 2bit quantization for kv cache. *arXiv preprint arXiv:2402.02750*, 2024b.

Alexander Ly, Maarten Marsman, Josine Verhagen, Raoul Grasman, and Eric-Jan Wagenmakers. A tutorial on fisher information, 2017.

Xinyin Ma, Gongfan Fang, and Xinchao Wang. Llm-pruner: On the structural pruning of large language models, 2023.

Stephen Merity, Caiming Xiong, James Bradbury, and Richard Socher. Pointer sentinel mixture models, 2016.

Carl Dean Meyer. *Matrix Analysis and Applied Linear Algebra*. SIAM, 2000.

Todor Mihaylov, Peter Clark, Tushar Khot, and Ashish Sabharwal. Can a suit of armor conduct electricity? a new dataset for open book question answering. *arXiv preprint arXiv:1809.02789*, 2018.

Matan Ben Noach and Yoav Goldberg. Compressing pre-trained language models by matrix decomposition. In Kam-Fai Wong, Kevin Knight, and Hua Wu (eds.), *Proceedings of the 1st Conference of the Asia-Pacific Chapter of the Association for Computational Linguistics and the 10th International Joint Conference on Natural Language Processing, AACL/IJCNLP 2020, Suzhou, China, December 4-7, 2020*, pp. 884–889. Association for Computational Linguistics, 2020. URL `https://aclanthology.org/2020.aacl-main.88/`.

Ofir Press, Noah Smith, and Mike Lewis. Train short, test long: Attention with linear biases enables input length extrapolation. In *International Conference on Learning Representations*, 2022. URL `https://openreview.net/forum?id=R8sQPpGCv0`.

Colin Raffel, Noam Shazeer, Adam Roberts, Katherine Lee, Sharan Narang, Michael Matena, Yanqi Zhou, Wei Li, and Peter J Liu. Exploring the limits of transfer learning with a unified text-to-text transformer. *Journal of machine learning research*, 21(140):1–67, 2020.

Luka Ribar, Ivan Chelombiev, Luke Hudlass-Galley, Charlie Blake, Carlo Luschi, and Douglas Orr. Sparq attention: Bandwidth-efficient LLM inference. In *ICLR 2024 Workshop on Mathematical and Empirical Understanding of Foundation Models*, 2024. URL `https://openreview.net/forum?id=Ue8EHzaFI4`.

Keisuke Sakaguchi, Ronan Le Bras, Chandra Bhagavatula, and Yejin Choi. Winogrande: An adversarial winograd schema challenge at scale. *Communications of the ACM*, 64(9):99–106, 2021.

Charbel Sakr and Brucek Khailany. Espace: Dimensionality reduction of activations for model compression. *arXiv preprint arXiv:2410.05437*, 2024.

Pratyusha Sharma, Jordan T. Ash, and Dipendra Misra. The truth is in there: Improving reasoning in language models with layer-selective rank reduction, 2023.

Noam Shazeer. Fast transformer decoding: One write-head is all you need, 2019. URL `https://arxiv.org/abs/1911.02150`.

Jianlin Su, Yu Lu, Shengfeng Pan, Bo Wen, and Yunfeng Liu. Roformer: Enhanced transformer with rotary position embedding. *CoRR*, abs/2104.09864, 2021. URL `https://arxiv.org/abs/2104.09864`.

Jiaming Tang, Yilong Zhao, Kan Zhu, Guangxuan Xiao, Baris Kasikci, and Song Han. QUEST: Query-Aware Sparsity for Efficient Long-Context LLM Inference. In *Proceedings of the International Conference on Machine Learning (ICML)*, 2024.

Philippe Tillet, Hsiang-Tsung Kung, and David Cox. Triton: an intermediate language and compiler for tiled neural network computations. In *Proceedings of the 3rd ACM SIGPLAN International Workshop on Machine Learning and Programming Languages*, pp. 10–19, 2019.

Hugo Touvron, Louis Martin, Kevin Stone, Peter Albert, Amjad Almahairi, Yasmine Babaei, Nikolay Bashlykov, Soumya Batra, Prajjwal Bhargava, Shruti Bhosale, Dan Bikel, Lukas Blecher, Cristian Canton Ferrer, Moya Chen, Guillem Cucurull, David Esiobu, Jude Fernandes, Jeremy Fu, Wenyin Fu, Brian Fuller, Cynthia Gao, Vedanuj Goswami, Naman Goyal, Anthony Hartshorn, Saghar Hosseini, Rui Hou, Hakan Inan, Marcin Kardas, Viktor Kerkez, Madian Khabsa, Isabel Kloumann, Artem Korenev, Punit Singh Koura, Marie-Anne Lachaux, Thibaut Lavril, Jenya Lee, Diana Liskovich, Yinghai Lu, Yuning Mao, Xavier Martinet, Todor Mihaylov, Pushkar Mishra, Igor Molybog, Yixin Nie, Andrew Poulton, Jeremy Reizenstein, Rashi Rungta, Kalyan Saladi, Alan Schelten, Ruan Silva, Eric Michael Smith, Ranjan Subramanian, Xiaoqing Ellen Tan, Binh Tang, Ross Taylor, Adina Williams, Jian Xiang Kuan, Puxin Xu, Zheng Yan, Iliyan Zarov, Yuchen Zhang, Angela Fan, Melanie Kambadur, Sharan Narang, Aurelien Rodriguez, Robert Stojnic, Sergey Edunov, and Thomas Scialom. Llama 2: Open foundation and fine-tuned chat models, 2023.

Albert Tseng, Jerry Chee, Qingyao Sun, Volodymyr Kuleshov, and Christopher De Sa. Quip#: Even better llm quantization with hadamard incoherence and lattice codebooks. *arXiv preprint arXiv:2402.04396*, 2024.

Ashish Vaswani, Noam Shazeer, Niki Parmar, Jakob Uszkoreit, Llion Jones, Aidan N. Gomez, Lukasz Kaiser, and Illia Polosukhin. Attention is all you need. In Isabelle Guyon, Ulrike von Luxburg, Samy Bengio, Hanna M. Wallach, Rob Fergus, S. V. N. Vishwanathan, and Roman Garnett (eds.), *Advances in Neural Information Processing Systems 30: Annual Conference on Neural Information Processing Systems 2017, December 4-9, 2017, Long Beach, CA, USA*, pp. 5998–6008, 2017. URL `https://proceedings.neurips.cc/paper/2017/hash/3f5ee243547dee91fbd053c1c4a845aa-Abstract.html`.

Xin Wang, Yu Zheng, Zhongwei Wan, and Mi Zhang. Svd-llm: Truncation-aware singular value decomposition for large language model compression. *arXiv preprint arXiv:2403.07378*, 2024.

Thomas Wolf, Lysandre Debut, Victor Sanh, Julien Chaumond, Clement Delangue, Anthony Moi, Pierric Cistac, Tim Rault, Remi Louf, Morgan Funtowicz, Joe Davison, Sam Shleifer, Patrick von Platen, Clara Ma, Yacine Jernite, Julien Plu, Canwen Xu, Teven Le Scao, Sylvain Gugger, Mariama Drame, Quentin Lhoest, and Alexander Rush. Transformers: State-of-the-art natural language processing. In Qun Liu and David Schlangen (eds.), *Proceedings of the 2020 Conference on Empirical Methods in Natural Language Processing: System Demonstrations*, pp. 38–45, Online, October 2020. Association for Computational Linguistics. doi: 10.18653/v1/2020. emnlp-demos.6. URL `https://aclanthology.org/2020.emnlp-demos.6`.

Guangxuan Xiao, Yuandong Tian, Beidi Chen, Song Han, and Mike Lewis. Efficient streaming language models with attention sinks. In *The Twelfth International Conference on Learning Representations*, 2024. URL `https://openreview.net/forum?id=NG7sS51zVF`.

Zhihang Yuan, Yuzhang Shang, Yue Song, Qiang Wu, Yan Yan, and Guangyu Sun. Asvd: Activation-aware singular value decomposition for compressing large language models, 2023.

Zhihang Yuan, Yuzhang Shang, Yang Zhou, Zhen Dong, Zhe Zhou, Chenhao Xue, Bingzhe Wu, Zhikai Li, Qingyi Gu, Yong Jae Lee, Yan Yan, Beidi Chen, Guangyu Sun, and Kurt Keutzer. LLM inference unveiled: Survey and roofline model insights. *CoRR*, abs/2402.16363, 2024. doi: 10.48550/ARXIV.2402.16363. URL `https://doi.org/10.48550/arXiv.2402. 16363`.

Yuxuan Yue, Zhihang Yuan, Haojie Duanmu, Sifan Zhou, Jianlong Wu, and Liqiang Nie. Wkvquant: Quantizing weight and key/value cache for large language models gains more. *arXiv preprint arXiv:2402.12065*, 2024.

Rowan Zellers, Ari Holtzman, Yonatan Bisk, Ali Farhadi, and Yejin Choi. Hellaswag: Can a machine really finish your sentence? *arXiv preprint arXiv:1905.07830*, 2019.

Zhenyu Zhang, Ying Sheng, Tianyi Zhou, Tianlong Chen, Lianmin Zheng, Ruisi Cai, Zhao Song, Yuandong Tian, Christopher Ré, Clark Barrett, et al. H2o: Heavy-hitter oracle for efficient generative inference of large language models. *Advances in Neural Information Processing Systems*, 36, 2024.

Jiawei Zhao, Zhenyu Zhang, Beidi Chen, Zhangyang Wang, Anima Anandkumar, and Yuandong Tian. Galore: Memory-efficient LLM training by gradient low-rank projection. *CoRR*, abs/2403.03507, 2024. doi: 10.48550/ARXIV.2403.03507. URL `https://doi.org/10. 48550/arXiv.2403.03507`.

Yilong Zhao, Chien-Yu Lin, Kan Zhu, Zihao Ye, Lequn Chen, Size Zhenga, Luis Ceze, Arvind Krishnamurthy, Tianqi Chen, and Baris Kasikci. Atom: Low-bit quantization for efficient and accurate llm serving. *arXiv preprint arXiv:2310.19102*, 2023.

Ming Zhong, Da Yin, Tao Yu, Ahmad Zaidi, Mutethia Mutuma, Rahul Jha, Ahmed Hassan Awadallah, Asli Celikyilmaz, Yang Liu, Xipeng Qiu, and Dragomir Radev. QMSum: A new benchmark for query-based multi-domain meeting summarization. In Kristina Toutanova, Anna Rumshisky, Luke Zettlemoyer, Dilek Hakkani-Tur, Iz Beltagy, Steven Bethard, Ryan Cotterell, Tanmoy Chakraborty, and Yichao Zhou (eds.), *Proceedings of the 2021 Conference of the North American Chapter of the Association for Computational Linguistics: Human Language Technologies*, pp. 5905–5921, Online, June 2021. Association for Computational Linguistics. doi: 10.18653/v1/ 2021.naacl-main.472. URL `https://aclanthology.org/2021.naacl-main.472`.

## APPENDIX

## A  QUANTIZATION BASICS

Quantization techniques use discrete low-bit values to approximate high-precision floating points. The general asymmetric uniform quantization function is defined as:

$$\overline{\mathbf{X}} = \text{clamp}\left(\left\lfloor \frac{\mathbf{X}}{s} \right\rceil + z, 0, 2^B - 1\right), \tag{9}$$

where $\overline{\mathbf{X}}$ denotes the approximated tensor with low-bit representations (i.e., 4-bit integers), $\mathbf{X}$ is the floating-point tensor, $s = \frac{\mathbf{X}_{\max} - \mathbf{X}_{\min}}{2^B - 1}$ is the scaling factor, and $z = \left\lfloor \frac{-\mathbf{X}_{\min}}{s} \right\rceil$ is a zero-point. The $\lfloor \cdot \rceil$ is the rounding operation.

## B  KERNEL IMPLEMENTATION DETAILS

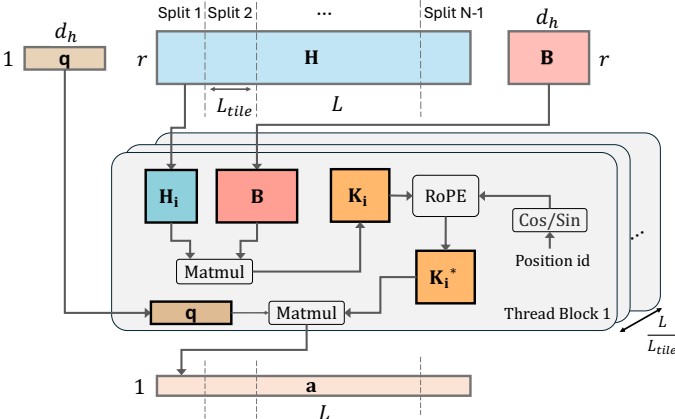

Figure 7: Illustration of our fused GPU kernel for computing attention scores with online reconstruction. In this figure, $\mathbf{q}$ represents the query vector, $\mathbf{H}$ denotes the low-rank compressed key states, and $\mathbf{B}$ stands for the reconstruction matrices.

**Kernel for attention score calculation with reconstruction.**    The central idea of *Palu* is to leverage low-rank latent representations to accelerate the attention mechanism by reducing data transfer overhead. Instead of working directly with the full-sized key matrix, we store and transfer a compressed low-rank latent representation, denoted as $\mathbf{H} \in \mathbb{R}^{L \times r}$. During computation, our custom GPU kernel performs an on-the-fly reconstruction using a reconstruction matrix $\mathbf{B} \in \mathbb{R}^{r \times d_h}$, producing a restored key matrix $\mathbf{K} \in \mathbb{R}^{L \times d_h}$, where $L$ is the sequence length, $d_h$ is the hidden dimension, and $r$ denote the remaining rank after performing low-rank projection. The query vector, represented as $\mathbf{q} \in \mathbb{R}^{1 \times d_h}$, then multiplies with the reconstructed keys to obtain the attention scores.

To efficiently leverage parallelism, we perform tiling along the sequence length dimension $L$. Specifically, we split the sequence into smaller tiles of size $L_{\text{tile}}$, assigning each tile to a dedicated thread block. Each thread block independently reconstructs a submatrix $\mathbf{H}_i \in \mathbb{R}^{L_{\text{tile}} \times d_h}$ from the low-rank latent representation $\mathbf{H}$, then applies the positional embedding using RoPE, and finally performs the matrix-vector multiplication between $\mathbf{q}$ and $\mathbf{H}_i$ to produce partial attention scores. This design ensures that all intermediate computations, from reconstruction to embedding and final multiplication, remain entirely in on-chip memory (*i.e.,* share memory), thus minimizing high-latency memory access and taking full advantage of the GPU's parallel processing capabilities to achieve significant speedups.

## C   DISCUSSION REGARDING MEMORY USAGE

In this work, the experimental results focus on the compression rate of the KV-Cache as a key metric. However, it is crucial to consider overall memory savings as a more significant factor. For instance, as demonstrated in Sec. 2.2, a typical compression rate of 30% can lead to an increase in weight size by approximately 40%. This increase is calculated under the assumption that $m = n$ and $r = 0.7n$, resulting in the equation $\frac{mr+nr}{mn} = 1.4$. Such an increase indicates substantial extra memory usage.

This issue primarily arises in J-LRD decomposition schemes, where the projections of all heads are decomposed jointly. In contrast, our M-LRD decomposition schemes and optimized G-LRD schemes involve non-square target matrices. For example, in the G-LRD scheme with a group size of 4, the target matrix is formed by concatenating the original projection matrices of each attention head in the group. In the Llama-2-7b model, with an embedding dimension of 4096 and head dimensions of 128, each projection matrix is 4096x128, resulting in a concatenated matrix of size 4096x512. In this case, the dimensions should be considered as $m = 8n$. Applying the referenced equation $\frac{mr+nr}{mn}$ with $r = 0.7n$, we find that $\frac{mr+nr}{mn} = 0.7875$, indicating no additional storage cost and, in fact, achieving an additional 21.25% memory savings.

Furthermore, it is important to highlight that the weights associated with the K and V projections account for only 2 out of 7 linear layers within transformer blocks, comprising merely 16% of the parameters in Llama-2-7b models. This limits the overall impact on memory usage. Thus, while J-LRD may incur overhead, the M-LRD and G-LRD schemes provide efficient alternatives that do not lead to increased memory usage, making them viable options for practical applications.

Table 4: Evaluation integrating LoRA with Palu on Llama2-7B.

| Comp. Rate | Method | Zero-Shot Accuracy (%) ↑ | | | | | | | |
| --- | --- | --- | --- | --- | --- | --- | --- | --- | --- |
| | | OBQA | HellaSwag | PIQA | ARC-e | ARC-c | WinoGrande | Avg. | Diff. |
| Rate = 0% | baseline | 44.20 | 76.00 | 78.07 | 76.30 | 46.42 | 69.30 | 65.05 | - |
| Rate = 50% w/o LoRA | J-LRD | 45.40 | 75.57 | 77.48 | 75.97 | 45.31 | 69.22 | 64.83 | -0.22 |
| | G-LRD | 43.60 | 73.39 | 76.33 | 73.02 | 42.57 | 66.77 | 62.61 | -2.44 |
| | M-LRD | 39.60 | 65.35 | 74.76 | 67.17 | 35.24 | 64.64 | 57.79 | -7.26 |
| Rate = 50% w/ LoRA | J-LRD | 44.20 | 74.09 | 78.51 | 77.27 | 48.81 | 71.03 | 65.65 | +0.60 |
| | G-LRD | 43.40 | 73.08 | 78.56 | 75.72 | 47.10 | 69.85 | 64.62 | -0.43 |
| | M-LRD | 41.80 | 70.78 | 78.02 | 73.86 | 43.86 | 69.22 | 62.92 | -2.12 |

## D   INTEGRATING *Palu* WITH LORA FINETUNE

LoRA (Hu et al., 2022) has become one of the most widely used efficient fine-tuning techniques for adapting models to particular tasks or domains with limited data. It has also been applied with LLM compression approaches (Wang et al., 2024; Ma et al., 2023) as a post-compression recovery technique to recover information loss after compression. In *Palu*, LoRA is also applicable to boost the accuracy further.

To integrate LoRA with *Palu*, we introduce additional low-rank matrices $\mathbf{A}'_r \in \mathbb{R}^{d \times r'}$ and $\mathbf{B}'_r \in \mathbb{R}^{r' \times d}$ to refine the original low-rank projection as below:

$$\mathbf{h} = \mathbf{Ax} + \mathbf{A_{r'}B_{r'}x} \tag{10}$$

Here, $\mathbf{A}$ will be fixed parameters derived from low-rank decomposition from pre-trained weights of linear layers, while $\mathbf{A}'_r$ and $\mathbf{B}'_r$ are trainable parameters to capture the task-specific nuances and recovers the information lost during the compression.

**Setup.**   Following Ashkboos et al. 2024a, we sample 8k samples from the Alpaca training dataset as a fine-tuning dataset and apply LoRA with rank $r' = 32$ and $\alpha = 32$. All other hyper-parameters are aligned with Ashkboos et al. (2024a), except for the learning rate $2e - 4$, and the use of a cosine learning rate scheduler.

Table 5: Perplexity and zero-shot accuracy of *Palu*, with different decomposition strategies at 50%

| Model | Method | Perplexity ↓ | | Zero-Shot Accuracy (%) ↑ | | | | | | |
|-------|--------|------|------|-------|-------|-------|-------|-------|-------|-------|
| | | Wiki2 | C4 | OBQA | Hella | PIQA | ARC-e | ARC-c | Wino | Avg. |
| Llama2-13B | Baseline | 4.88 | 6.70 | 45.20 | 79.39 | 79.11 | 79.42 | 49.06 | 72.38 | 67.43 |
| | J-LRD | 4.97 | 6.92 | 46.40 | 79.48 | 78.62 | 79.29 | 49.91 | 70.56 | 67.38 |
| | G-LRD | 5.31 | 7.76 | 45.60 | 77.29 | 77.42 | 76.05 | 45.99 | 72.45 | 65.80 |
| | M-LRD | 5.65 | 8.34 | 43.20 | 74.34 | 77.53 | 75.76 | 45.39 | 68.98 | 64.20 |

**Experiment Results.**  We present the experiment results with LoRA in Table 4. Following Ashkboos et al. 2024a With LoRA incorporated, J-LRD continues to show minimal performance degradation with an average drop of 1.00%. G-LRD (gs=4) and M-LRD show improved results compared to their non-LoRA counterparts, with average drops of 2.01% and 5.14%, respectively. Notably, with LoRA integration, G-LRD shows only a 1.03% accuracy difference compared to J-LRD.

# E   MORE RESULTS ON ZERO-SHOT ACCURACY

Following up Sec. 4.2, we further report the perplexity and zero-shot evaluation results of *Palu* on the Llama-2-13B at 50% compression rate. As shown in Table. 5, we observe that *Palu* achieve competitive accuracy drops around 3% or less across different using either J-LRD, G-LRD, or M-LRD. Thus, the users may adopt M-LRD first to optimize the efficiency further.

Table 6: Ablation study of low-rank decomposition group size on perplexity for the Llama2-7B model at a 50% compression rate using Wikitext-2.

| Method | Group Size | Perplexity |
|--------|-----------|------------|
| Baseline | - | 5.47 |
| J-LRD | 32 | 5.62 |
| G-LRD | 16 | 5.74 |
| | 8 | 5.88 |
| | 4 | 6.01 |
| | 2 | 6.42 |
| M-LRD | 1 | 6.81 |

# F   ABLATION STUDY

## F.1   INFLUENCE OF DIFFERENT GROUP SIZE

Since our proposed G-LRD method allows for balancing performance and efficiency by adjusting the group size, we conducted an ablation study on group size. As seen in Table 6, as the group size increases, the amount of shared information also increases, leading to improved performance.

## F.2   INFLUENCE OF WALSH-HADAMARD TRANSFORM

We conduct the ablation study to profile the benefits of applying the Walsh-Hadamard Transform (WHT). Experiment results are reported at Table 7. On the 3-bit quantization level, we observe that the Hadamard Transform only brings a slight amount of perplexity. However, when we quantize the low-rank representation more extremely (*i.e.,* 2-bit), we can observe a notable 4.17 perplexity enhancements. It's worth re-emphasizing that Hadamard Transform will not bring extra overhead during inference, as *Palu* optimizes the WHT process via offline preprocessing. The reader may refer to Sec. 3.4 for more details.

## F.3   AUTOMATIC RANK ALLOCATION VS. UNIFORM RANK ALLOCATION

Table 8 presents the ablation study on the impact of different rank allocation schemes on the model's accuracy. Applying rank searching results in a notable performance improvement. For instance, at a

compression rate of 50%, there is a significant reduction in perplexity by 2.18. Fig. 8 visualizes the rank allocation across different transformer blocks for key and value projection layers. The results clearly demonstrate a non-uniform allocation result. Specifically, we observe that the value is generally allocated a higher rank than the key. Additionally, the first half of the layers are assigned higher ranks, indicating their greater importance in preserving model performance. This visualization underscores the effectiveness of our rank search algorithm in identifying and allocating appropriate ranks to different components, thereby optimizing the balance between compression and accuracy.

Table 7: Ablation Study on different quantization settings for quantizing low-rank latent representations. Same as Sec. 4.3, we use the WikiText-2 with sequence length set to 4096 as the evaluation benchmark.

| Method | Wikitext-2 PPL ↓ |
|---|---|
| Llama2-7B | 5.12 |
| Palu-30% (FP16) | 5.25 |
| + 3-bits w/o Hadamard | 5.52 |
| + 3-bits w Hadamard | 5.33 (0.19↓) |
| + 2-bits w/o Hadamard | 9.48 |
| + 2-bits w Hadamard | 5.77 (3.71↓) |
| Palu-50% (FP16) | 5.63 |
| + 3-bits w/o Hadamard | 5.99 |
| + 3-bits w Hadamard | 5.77 (0.22↓) |
| + 2-bits w/o Hadamard | 10.58 |
| + 2-bits w Hadamard | 6.41 (4.17↓) |

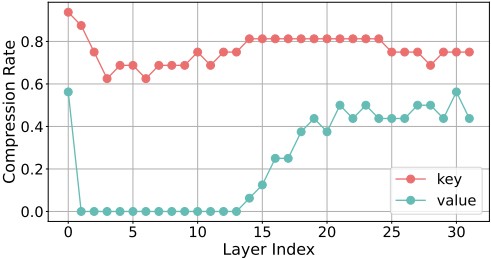

Figure 8: Visualization of layer-wise low-rank compression rate on Llama-2-7B with 50% of overall compression rate. Here, compression rates (*i.e.,* rank) are allocated using the proposed Fisher Information-based automated rank allocation algorithm.

Table 8: Ablation study on w/ and w/o rank search. We use Llama2-7b and Wikitext-2 with sequence length 2048 as the benchmark.

| | Rate=30% | Rate=50% | Rate=70% |
|---|---|---|---|
| Uniform | 6.34 | 7.36 | 10.77 |
| Automatic (ours) | 5.62 (0.72↓) | 6.02 (1.36↓) | 8.59 (2.18↓) |

# G EXPERIMENT DETAILS

## G.1 ZERO-SHOT EVALUATION DETAILS

We selected six zero-shot tasks from the LM-eval benchmark to evaluate *Palu*:

- OpenBookQA (accuracy, Mihaylov et al.)

- HellaSwag (acc_norm, Zellers et al.)
- PIQA (accuracy, Bisk et al.)
- ARC-Easy (accuracy, Clark et al.)
- ARC-Challenge (acc_norm, Clark et al.)
- WinoGrande (accuracy, Sakaguchi et al.)

We report accuracy for WinoGrande, PIQA, and ARC-Easy, and accuracy normalized by sequence length (acc_norm) for HellaSwag and ARC-Challenge.

## G.2 LongBench Evaluation Details

For the LongBench evaluation in this manuscript, we conducted tests on all available English tasks. These comprise sixteen tasks categorized into six subgroups, ensuring a comprehensive evaluation of *Palu*. The tasks and their corresponding metrics are detailed below:

- Single-Document QA:
  - Qasper (F1 score, Dasigi et al.)
  - NarrativeQA (F1 score)
  - MultiFieldQA-en (F1 score)
- Multi-Document QA:
  - HotpotQA (F1 score, Dasigi et al.)
  - 2WikiMultihopQA (F1 score)
  - MuSiQue (F1 score)
- Summarization:
  - QMSum (ROUGE score, Zhong et al.)
  - MultiNews (ROUGE score, Fabbri et al.)
  - GovReport (ROUGE score)
- Few-shot Learning:
  - TREC (classification score, Li & Roth)
  - TriviaQA (F1 score, Joshi et al.)
  - SAMSum (ROUGE score, Gliwa et al.)
- Code Completion:
  - LCC (similarity score, Guo et al.)
  - RepoBench-P (similarity score, Liu et al.)
- Synthetic:
  - PassageCount (Accuracy)
  - PassageRetrevial (Accuracy)

During the evaluation, we set the maximum sequence length to 31500 for both the Mistral and LongChat model.

## H   Discussion of Directly Performing SVD on KV-Cache During Runtime

As discussed at the beginning of Sec. 3.1, a straightforward approach to compress the KV-Cache with low-rank projections is to apply Singular Value Decomposition (SVD) directly to the KV-Cache. To evaluate its feasibility, we compare the latency required for performing SVD to the time taken for a forward pass through a decoder block, as illustrated in Tab.9. The results clearly demonstrate that performing SVD on the fly introduces significant computational overhead. Specifically, runtime SVD is approximately 5–10× slower than a single forward pass of the decoder block.

| Seqlen | 32k | 64k | 128k |
|---|---|---|---|
| Decoder Block (fp16) | 0.231s | 0.65s | 1.93s |
| SVD | 9.24s | 9.90s | 11.44s |

Table 9: Latency comparison between performing SVD on the KV-Cache and a single forward pass through the transformer decoder block, using the model configuration of Llama-2-7B

Given these findings, we adopt static weight decomposition techniques and modify the caching mechanism to store the lower-dimensional latent representations, thereby reducing the memory footprint while avoiding the runtime costs associated with on-the-fly SVD.

## I  MORE RELATED WORK

**Token Eviction.**   One prominent direction for reducing the memory footprint of KV-Cache is KV-Cache eviction Adnan et al. (2024); Ge et al. (2024); Xiao et al. (2024); Zhang et al. (2024). KV-Cache eviction techniques selectively retain parts of the KV-Cache and discard less important tokens to maintain the use of a fixed-size KV-Cache to control memory usage. Representative works, such as AttentionSink (Xiao et al., 2024), employ a fixed eviction policy by preserving the tokens in the very beginning, which is also called attention sink, together with recent KV pairs. H2O (Zhang et al., 2024) selected the tokens based on accumulative attention scores. SnapKV (Li et al., 2024) evicts non-important tokens of each head based on the local observation window of prompts. While these methods reduce the memory footprint of KV-Cache and data to be transferred, they permanently discard KV pairs deemed less important, leading to accuracy degradation in some complex tasks that may require information from those eviction parts of the sequence.

**Token Selection.**   To address the limitations of token eviction methods, another line of research focuses on retaining the entire KV-Cache while employing sparse attention mechanisms to process only selected parts of the KV-Cache, thereby reducing latency. Notable examples include SparQ (Ribar et al., 2024) and Quest (Tang et al., 2024). These methods achieve significant improvements in latency and accuracy preservation. However, since the full KV-Cache is retained, memory requirements remain unaddressed. As a result, techniques like CPU offloading are often necessary to execute inference requests.

In *Palu*, we take an orthogonal approach by compressing the hidden dimensions of the KV-Cache via low-rank projections. A potential direction for future work could be combining *Palu* with token selection methods to reduce memory usage further. This approach could involve compressing non-salient or infrequently accessed tokens to enhance memory efficiency and overall system performance further.

## J  DISCUSSION OF ALTERNATIVE STRATEGIES FOR DERIVING LOW-RANK PROJECTION MATRICES

In Palu, low-rank matrices $\mathbf{W} \approx \mathbf{AB}$ are derived by applying Singular Value Decomposition (SVD) to the weight matrices. Aside from this approach, alternative strategies (Chen et al., 2021; Sakr & Khailany, 2024) for obtaining low-rank projection matrices also exist. For instance, ESPACE Sakr & Khailany (2024) performs calibration on the input $\mathbf{x}$ and subsequently applies eigenvector decomposition to compute the low-rank projection matrix $\mathbf{P}$. During inference, ESPACE operates as $\mathbf{h} = \mathbf{xA}'$ and $\mathbf{y} = \mathbf{hB}'$, where $\mathbf{A}' = \mathbf{P}$ and $\mathbf{B}' = \mathbf{WP}$ are pre-computed. This approach ultimately achieves the same down-projection and reconstruction mechanism in the forward pass as the weight decomposition method used in *Palu*.

Future research could investigate accuracy differences between deriving low-rank projection matrices through direct weight decomposition, as implemented in *Palu*, and alternative techniques like ESPACE. Additionally, *Palu*'s GPU kernel optimizations and quantization-friendly enhancements—such as the fusion of the Hadamard Transform—are agnostic to the method used to derive

low-rank projection matrices. This adaptability provides a robust framework for integrating ES-PACE or other approaches in future work.

# K SVD DECOMPOSITION DETAILS AND ERROR BOUND ANALYSIS

In the proposed *Palu* KV-Cache compression framework, we perform low-rank decomposition onto the Key and Value projection matrices of each layer. The output error introduced by this decomposition directly corresponds to the error induced in the KV-Cache. To minimize this error, we employ the truncation-aware SVD technique introduced by (Wang et al., 2024). This method enhances the standard SVD approach by incorporating a transformation that accounts for the statistical properties of the activation data, thereby reducing the output error caused by decomposition. Below, we outline the algorithm details and provide an error analysis for the resulting KV-Cache output error, demonstrating that the introduced error can be bounded.

## K.1 TRUNCATION-AWARE SVD WITH WHITENING TRANSFORMATION

**1. Weight Transformation.** The primary idea is to adjust the weight matrix $\mathbf{W} \in \mathbb{R}^{m \times n}$ by introducing an invertible transformation matrix $\mathbf{S} \in \mathbb{R}^{m \times m}$. This transformation aligns $\mathbf{W}$ with the activation data $\mathbf{X} \in \mathbb{R}^{b \times m}$, making the subsequent SVD more effective in capturing the essential components.

We express the transformed output as:

$$\mathbf{XW} = \mathbf{XS}^{-1}\mathbf{SW} = \tilde{\mathbf{X}}\tilde{\mathbf{W}},$$

where

$$\tilde{\mathbf{X}} = \mathbf{XS}^{-1} \in \mathbb{R}^{b \times m}, \quad \tilde{\mathbf{W}} = \mathbf{SW} \in \mathbb{R}^{m \times n}.$$

Following Wang et al. (2024), the transformation matrix $\mathbf{S}$ is derived using the Cholesky decomposition (Meyer, 2000) of the covariance matrix of the activation data:

$$\mathbf{SS}^\top = \mathbf{X}^\top \mathbf{X} + \lambda \mathbf{I},$$

with $\lambda > 0$ being a small regularization parameter to ensure numerical stability, and $\mathbf{I}$ is the identity matrix. This choice of $\mathbf{S}$ ensures that the transformation accounts for the correlations in the input activations, thereby minimizing the output error introduced by the decomposition.

**2. SVD on Post-transformed Weights.** Following the transformation, we perform standard Singular Value Decomposition (SVD) on the transformed weight matrix $\tilde{\mathbf{W}}$:

$$\tilde{\mathbf{W}} = \mathbf{U}\boldsymbol{\Sigma}\mathbf{V}^\top$$

**3. Deriving Low-Rank Matrices.** To achieve compression, we retain the top $r$ singular values and their corresponding singular vectors, truncating the remaining $s = t - r$ singular values:

$$\boldsymbol{\Sigma}_r = \mathrm{diag}(\sigma_1, \sigma_2, \ldots, \sigma_r), \quad \mathbf{U}_r \in \mathbb{R}^{m \times r}, \quad \mathbf{V}_r \in \mathbb{R}^{n \times r},$$

where $t = \min(m, n)$ represents the number of singular values in the decomposition. The compressed weight matrix $\mathbf{W}'$ is then reconstructed by reversing the transformation:

$$\mathbf{W}' = \mathbf{S}^{-1}\mathbf{U}_r\boldsymbol{\Sigma}_r\mathbf{V}_r^\top.$$

This can be expressed as the product of two low-rank matrices $\mathbf{A}$ and $\mathbf{B}$:

$$\mathbf{W}' = \mathbf{AB}, \quad \text{where} \quad \mathbf{A} = \mathbf{S}^{-1}\mathbf{U}_r\sqrt{\boldsymbol{\Sigma}_r} \in \mathbb{R}^{m \times r}, \quad \mathbf{B} = \sqrt{\boldsymbol{\Sigma}_r}\mathbf{V}_r^\top \in \mathbb{R}^{r \times n}.$$

## K.2 ERROR BOUND ANALYSIS

We now analyze the error bound of the output error resulting from the compression. The analysis is divided into two parts:

**1. Expressing the Output Error in Singular Values.** Let the output of the original linear layer and low-rank approximated counterpart to be $\mathbf{Y}$ and $\mathbf{Y}'$, individually,

The output error is defined as:

$$\mathbf{Y} - \mathbf{Y}' = \mathbf{X}\mathbf{W} - \mathbf{X}\mathbf{W}' = \tilde{\mathbf{X}}\tilde{\mathbf{W}} - \tilde{\mathbf{X}}\tilde{\mathbf{W}}' = \tilde{\mathbf{X}}(\tilde{\mathbf{W}} - \tilde{\mathbf{W}}').$$

Substituting the SVD decompositions:

$$\tilde{\mathbf{W}} - \tilde{\mathbf{W}}' = \mathbf{U}\boldsymbol{\Sigma}\mathbf{V}^\top - \mathbf{U}_r\boldsymbol{\Sigma}_r\mathbf{V}_r^\top = \mathbf{U}[:, r+1:t]\boldsymbol{\Sigma}_T\mathbf{V}[:, r+1:t]^\top,$$

where $\boldsymbol{\Sigma}_T = \mathrm{diag}(\sigma_{r+1}, \ldots, \sigma_t)$ contains the truncated singular values, and $\mathbf{U}[:, r+1:t]$ and $\mathbf{V}[:, r+1:t]$ denote the columns corresponding to the truncated singular vectors.

Therefore, the output error becomes:

$$\mathbf{Y} - \mathbf{Y}' = \tilde{\mathbf{X}}\mathbf{U}[:, r+1:t]\boldsymbol{\Sigma}_T\mathbf{V}[:, r+1:t]^\top.$$

Assuming that $\tilde{\mathbf{X}}^\top\tilde{\mathbf{X}} \approx \mathbf{I}$ (which holds when $\lambda$ is small), the squared Frobenius norm of the output error is:

$$\begin{aligned}
\|\mathbf{Y} - \mathbf{Y}'\|_F^2 &= \mathrm{Tr}\left((\mathbf{Y} - \mathbf{Y}')^\top(\mathbf{Y} - \mathbf{Y}')\right) \\
&= \mathrm{Tr}\left((\boldsymbol{\Sigma}_T\mathbf{V}_T^\top)^\top\mathbf{U}_T^\top\tilde{\mathbf{X}}^\top\tilde{\mathbf{X}}\mathbf{U}_T\boldsymbol{\Sigma}_T\mathbf{V}_T^\top\right) \\
&\approx \mathrm{Tr}\left(\boldsymbol{\Sigma}_T^\top\boldsymbol{\Sigma}_T\right) = \sum_{i=r+1}^t \sigma_i^2.
\end{aligned}$$

Here, $\mathbf{U}_T = \mathbf{U}[:, r+1:t]$, $\mathbf{V}_T = \mathbf{V}[:, r+1:t]$ and $\mathrm{Tr}(\cdot)$ refer to matrix trace.

**2. Bounding the Singular Values.** Each truncated singular value $\sigma_i$ satisfies:

$$\sigma_i \le \sigma_1 = \sigma_{\max}(\tilde{\mathbf{W}}) = \sigma_{\max}(\mathbf{S}\mathbf{W}).$$

The largest singular value of $\mathbf{S}\mathbf{W}$ can be bounded by:

$$\sigma_{\max}(\mathbf{S}\mathbf{W}) \le \|\mathbf{S}\|_2\|\mathbf{W}\|_2.$$

Since $\mathbf{S}$ is obtained via the Cholesky decomposition of $\mathbf{C} = \mathbf{X}^\top\mathbf{X} + \lambda\mathbf{I}$, its largest singular value is:

$$\|\mathbf{S}\|_2 = \sqrt{\sigma_{\max}(\mathbf{C})} = \sqrt{\sigma_{\max}(\mathbf{X}^\top\mathbf{X}) + \lambda}.$$

Therefore, the largest singular value $\sigma_{\max}(\mathbf{S}\mathbf{W})$ is bounded by:

$$\sigma_{\max}(\mathbf{S}\mathbf{W}) \le \sqrt{\sigma_{\max}(\mathbf{X}^\top\mathbf{X}) + \lambda}\,\|\mathbf{W}\|_2.$$

Combining these results, the Frobenius norm of the output error is bounded by:

$$\|\mathbf{Y} - \mathbf{Y}'\|_F \le \left(\sum_{i=r+1}^t \sigma_i^2\right)^{1/2} \le \sqrt{s}\,\sigma_{\max}(\mathbf{S}\mathbf{W}),$$

where $s = t - r$ is the number of truncated singular values.

Thus, the output error is bounded by:

$$\|\mathbf{Y} - \mathbf{Y}'\|_F \le \sqrt{s}\,\sqrt{\sigma_{\max}(\mathbf{X}^\top\mathbf{X}) + \lambda}\,\|\mathbf{W}\|_2 \le \sqrt{t}\,\sqrt{\sigma_{\max}(\mathbf{X}^\top\mathbf{X}) + \lambda}\,\|\mathbf{W}\|_2.$$

### K.3 SUMMARY

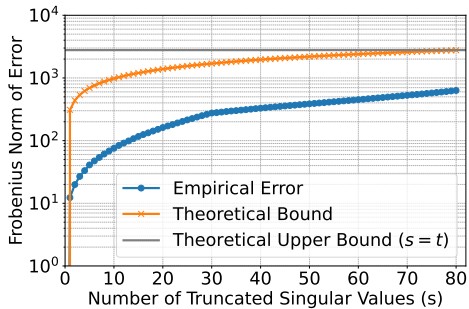

By applying whitening-based SVD compression, we obtain a low-rank approximation of $\mathbf{W}$ in the form $\mathbf{W} \approx \mathbf{AB}$, consistent with the standard SVD approach outlined in Section 2.2. Our theoretical analysis demonstrates that the output error of the decomposed linear layer can be effectively bounded and is influenced by both the number of truncated singular values and the properties of $\mathbf{X}$ and $\mathbf{W}$. Furthermore, we empirically validate these theoretical error bounds using randomly generated data ($m = 100$, $n = 80$, $b = 50$). As demonstrated in Fig. 9, empirical results confirm that the observed errors are consistently bounded by predicted bounds, which increase proportionally to the square root of the number of truncated singular values ($\sqrt{s}$).

Figure 9: Empirical and theoretical error bounds for the output of a low-rank decomposed linear layer with respect to the number of truncated ranks (singular values $s$)

## L  COMPUTE AND MEMORY FOOTPRINT ANALYSIS FOR ATTENTION SCORE COMPUTATION

Table 10: Summary of Complexity for Compute and Memory Requirements.

| Configuration | Compute Complexity | Memory Complexity |
|---|---|---|
| Baseline | $\mathcal{O}(L \cdot d_h \cdot n)$ | $\mathcal{O}(L \cdot d_h \cdot n)$ |
| Palu (M-LRD) | $\mathcal{O}(L \cdot r_i \cdot d_h \cdot n)$ | $\mathcal{O}(L \cdot r_i \cdot n)$ |
| Palu (G-LRD) | $\mathcal{O}(L \cdot r_g \cdot d_h \cdot n)$ | $\mathcal{O}(L \cdot r_g \cdot n_g)$ |
| Palu (J-LRD) | $\mathcal{O}(L \cdot r_{\text{joint}} \cdot d_h \cdot n)$ | $\mathcal{O}(L \cdot r_{\text{joint}})$ |

This section provides a detailed analysis of the compute and memory requirements for attention score computation in a RoPE-based attention module with a baseline approach and the *Palu*. The analysis considers various decomposition granularities, including M-LRD, G-LRD, and J-LRD, explicitly accounting for reconstruction, positional embedding (e.g., RoPE) re-application, and attention score computations (GEMV). We summarize the memory and compute complexity in Table. 10.

### L.1  DERIVATIONS

**General Formulation for Compute Complexity:**  The total FLOPs differ between the baseline and *Palu* configurations. For the baseline:

$$F_{\text{total}}^{\text{Baseline}} = F_{\text{GEMV}}.$$

For *Palu* configurations:

$$F_{\text{total}}^{Palu} = F_{\text{recons}} + F_{\text{RoPE}} + F_{\text{GEMV}}.$$

Here:

- $F_{\text{recons}}$: FLOPs to reconstruct the full key matrix $\mathbf{K}$ from low-rank latent representations.
- $F_{\text{RoPE}}$: FLOPs to re-apply positional embeddings onto $\mathbf{K}$.
- $F_{\text{GEMV}}$: FLOPs to compute attention scores via GEMV between $\mathbf{K}$ and the query $\mathbf{q}$.

**General Formulation for Memory Complexity:**  The total memory complexity includes both memory reads and writes:

$$M_{\text{total}} = M_{\text{reads}} + M_{\text{writes}},$$

where:

$$M_{\text{reads}} = M_{\text{keys}} + M_B + M_q, \quad M_{\text{writes}} = M_{\text{output}}.$$

Memory terms:

- $M_{\text{keys}}$: Memory to read the full keys in the baseline, or low-rank latents for *Palu*.

- $M_B$: Memory to read the reconstruction matrices.

- $M_q$: Memory to read the query vectors.

- $M_{\text{output}}$: Memory to write the computed attention scores.

**Baseline:** In the baseline configuration, the post-RoPE Key $\mathbf{K} \in \mathbb{R}^{L \times d_h}$ and query vector $\mathbf{q} \in \mathbb{R}^{1 \times d_h}$ are directly used for GEMV computations. Reconstruction and RoPE re-application are not required.

Compute complexity:

$$F_{\text{total}}^{\text{Baseline}} = L \cdot d_h \cdot n,$$
$$F_{\text{total}}^{\text{Baseline}} \approx \mathcal{O}(L \cdot d_h \cdot n).$$

Memory complexity:

$$M_{\text{total}}^{\text{Baseline}} = M_{\text{keys}}^{\text{Baseline}} + M_q + M_{\text{output}}$$
$$= (L \cdot d_h \cdot n) + (d_h \cdot n) + (L \cdot n),$$
$$M_{\text{total}}^{\text{Baseline}} \approx \mathcal{O}(L \cdot d_h \cdot n).$$

**Palu (M-LRD):** In the M-LRD configuration, each attention head has its own low-rank latent $\mathbf{H}_i \in \mathbb{R}^{L \times r_i}$ and reconstruction matrix $\mathbf{B}_i \in \mathbb{R}^{r_i \times d_h}$. Reconstruction and RoPE are required.

Compute complexity:

$$F_{\text{total}}^{\text{M-LRD}} = F_{\text{recons}}^{\text{M-LRD}} + F_{\text{RoPE}} + F_{\text{GEMV}}$$
$$= (L \cdot r_i \cdot d_h \cdot n) + (L \cdot d_h \cdot n) + (L \cdot d_h \cdot n),$$
$$= \left( L \cdot r_i \cdot d_h + 2 \cdot L \cdot d_h \right) \cdot n,$$
$$F_{\text{total}}^{\text{M-LRD}} \approx \mathcal{O}(L \cdot r_i \cdot d_h \cdot n).$$

Memory complexity:

$$M_{\text{total}}^{\text{M-LRD}} = M_{\text{keys}}^{\text{M-LRD}} + M_B^{\text{M-LRD}} + M_q + M_{\text{output}}$$
$$= (L \cdot r_i \cdot n) + (r_i \cdot d_h \cdot n) + (d_h \cdot n) + (L \cdot n),$$
$$\approx \mathcal{O}((L \cdot r_i \cdot n) + (r_i \cdot d_h \cdot n))$$
$$\approx \mathcal{O}(L \cdot r_i \cdot n) \qquad \text{when } L \gg d_h$$

**Palu (G-LRD):** In the G-LRD configuration, groups of $s$ heads share a low-rank latent $\mathbf{H}_{g_j} \in \mathbb{R}^{L \times r_g}$ and reconstruction matrix $\mathbf{B}_{g_j} \in \mathbb{R}^{r_g \times (d_h \cdot s)}$.

Compute complexity:

$$F_{\text{total}}^{\text{G-LRD}} = F_{\text{recons}}^{\text{G-LRD}} + F_{\text{RoPE}} + F_{\text{GEMV}}$$
$$= (L \cdot r_g \cdot d_h \cdot n) + (L \cdot d_h \cdot n) + (L \cdot d_h \cdot n),$$
$$= \left( L \cdot r_g \cdot d_h + 2 \cdot L \cdot d_h \right) \cdot n,$$
$$F_{\text{total}}^{\text{G-LRD}} \approx \mathcal{O}(L \cdot r_g \cdot d_h \cdot n).$$

Memory complexity:

$$M_{\text{total}}^{\text{G-LRD}} = M_{\text{keys}}^{\text{G-LRD}} + M_B^{\text{G-LRD}} + M_q + M_{\text{output}}$$
$$= (L \cdot r_g \cdot n_g) + (r_g \cdot d_h \cdot n) + (d_h \cdot n) + (L \cdot n),$$
$$\approx \mathcal{O}((L \cdot r_g \cdot n_g) + (r_g \cdot d_h \cdot n))$$
$$\approx \mathcal{O}((L \cdot r_g \cdot n_g)), \qquad \text{when } L \gg (d_h \cdot n)$$

**Palu (J-LRD):** In the J-LRD configuration, all $n$ heads share a single low-rank latent $\mathbf{H}_{\text{joint}} \in \mathbb{R}^{L \times r_{\text{joint}}}$ and reconstruction matrix $\mathbf{B}_{\text{joint}} \in \mathbb{R}^{r_{\text{joint}} \times (d_h \cdot n)}$.

Compute complexity:

$$
\begin{aligned}
F_{\text{total}}^{\text{J-LRD}} &= F_{\text{recons}}^{\text{J-LRD}} + F_{\text{RoPE}} + F_{\text{GEMV}} \\
&= (L \cdot r_{\text{joint}} \cdot d_h \cdot n) + (L \cdot d_h \cdot n) + (L \cdot d_h \cdot n), \\
&= \left(L \cdot r_{\text{joint}} \cdot d_h + 2 \cdot L \cdot d_h\right) \cdot n, \\
&\approx \mathcal{O}(L \cdot r_{\text{joint}} \cdot d_h \cdot n).
\end{aligned}
$$

Memory complexity:

$$
\begin{aligned}
M_{\text{total}}^{\text{J-LRD}} &= M_{\text{keys}}^{\text{J-LRD}} + M_B^{\text{J-LRD}} + M_q + M_{\text{output}} \\
&= (L \cdot r_{\text{joint}}) + (r_{\text{joint}} \cdot d_h \cdot n) + (d_h \cdot n) + (L \cdot n), \\
&\approx \mathcal{O}((L \cdot r_{\text{joint}}) + (r_{\text{joint}} \cdot d_h \cdot n)). \\
&\approx \mathcal{O}(L \cdot r_{\text{joint}}), \qquad \text{when } L \gg (d_h \cdot n)
\end{aligned}
$$

