# OpenReview forum: "Palu: KV-Cache Compression with Low-Rank Projection"
_ICLR.cc/2025/Conference — ICLR 2025 Poster_

### Official Review · Reviewer_pJHR · 2024-10-30

**Soundness:** 2
**Presentation:** 3
**Contribution:** 1
**Rating:** 5
**Confidence:** 4

**Summary:**

Palu is a new framework for compressing the key-value cache in large language models, reducing memory usage and improving inference speed. It uses low-rank projection, group-head low-rank decomposition, and automatic rank allocation to achieve up to 50% KV-cache compression with minimal accuracy loss. Palu also integrates quantization compatibility and optimized GPU kernels, resulting in significant speedups, especially for long-context scenarios. The framework outperforms state-of-the-art quantization methods and demonstrates effectiveness across various LLMs and benchmarks.

**Strengths:**

- Novelty: unlike common quantization and pruning approaches, Palu uses low rank decomposition to lower the size of KV caches, which is more mathematically sound and doesn't require hardware changes.
- Significant compression is claimed, and it is to be expected that KV caches can be significantly compressed. However, I do have some concerns about the achieved compression which I cover in the weaknesses.
- Extensive empirical results spanning accuracy studies and speed-up analyses are provided.

**Weaknesses:**

- While caching latent keys and values makes sense from the memory efficiency angle, it introduces significant computational overheads during decoding. Indeed, while the prefill phase can allow for GEMMs to occur at every layer, the decoding phase is all about GEMVs. However, in the palu framework, a full GEMM is required for to rematerialize each key and value matrix to be used in the attention module. This adds a lot of complexity, particularly when decoding a large output sequence length.
- What the paper really is doing is not KV cache compression, but really it is compressing the QKV WxA layer preceding attention by applying SVD on two thirds of the weight matrix.
- The experimental setup doesn't make a lot of sense. It is mentioned that FlashAttention is not used for apples-to-apples comparison. But FlashAttention anyway is there to speed-up the prefill phase where the KV cache doesn't exist yet. Why does it then matter? Why are you including prefill latency when your solution is supposed to be addressing KV cache compression, which is used in the decoding phase?

**Questions:**

- In Table 1, why does Llama2-7B have better perplexity and accuracy than Llama3-8B-Inst? There is definitely something wrong in the experimental setup.
- Rather than compressing weights, since KV cache compression is desired, can a method such as ESPACE [1] be used to project activations to a low rank directly?
[1] Sakr, Charbel, and Brucek Khailany. "ESPACE: Dimensionality Reduction of Activations for Model Compression." arXiv preprint arXiv:2410.05437 (2024).

---

> ### Author Response · Authors · 2024-11-19
>
> > (W1):  In the Palu framework, a full GEMM is required to rematerialize each key and value matrix to be used in the attention module. This adds a lot of complexity, particularly when decoding a large output sequence length.
>
> **Ans:** We acknowledge that computational overhead is a critical challenge when replacing full-dimensional KV-Cache with low-rank latent representations. However, we respectfully disagree with the characterization of this as a weakness. Addressing this challenge is central to Palu’s design, and we have proposed techniques such as G-LRD and custom GPU kernels to optimize the tradeoff between computation and memory efficiency. These innovations enable Palu to significantly reduce memory usage while accelerating LLM decoding. For a detailed explanation, we encourage the reviewer to refer to our global response.
>
> > (W2). What the paper is really doing is not KV cache compression, but really compressing the QKV WxA layer preceding attention.
>
> **Ans:** We respectfully disagree with the reviewer’s statement, as it mischaracterizes the focus of our work. While Palu achieves KV-Cache hidden-dimension reduction by decomposing the weight matrices of the Key and Value layers, its design and objectives are fundamentally different from weight compression. If the goal were weight compression, we could store KV-Cache in the original hidden dimension, eliminating the need for runtime reconstruction. However, Palu specifically targets KV-Cache compression, which necessitates caching latent tensors and dynamically reconstructing them during inference - a process far more complex than static weight compression.
>
> To address these challenges, Palu incorporates advanced techniques such as G-LRD, rank search, quantization integration, and custom GPU kernels to balance memory reduction and computational overhead effectively. These design choices are intrinsic to KV-Cache compression and not applicable to weight compression alone. We encourage the reviewer to refer to our global response for a detailed explanation of Palu’s unique focus and design.
>
>
> > (W3) The experimental setup doesn't make a lot of sense. It is mentioned that FlashAttention is not used for apples-to-apples comparisons. But FlashAttention anyway is there to speed up the prefill phase where the KV cache doesn't exist yet. Why does it then matter? Why are you including prefill latency when your solution is supposed to be addressing KV cache compression, which is used in the decoding phase?
>
> **Ans:** The latency results we reported in our paper are the decoding latency, not the prefill latency, as the reviewer pointed out that KV-Cache is used in the decoding stage. We have updated our manuscript to further clarify this part, which you can view in the revised submission.

---

> ### Author Response · Authors · 2024-11-19
>
> > (Q1) In Table 1, why does Llama2-7B have better perplexity and accuracy than Llama3-8B-Inst? There is definitely something wrong with the experimental setup.
>
> **Ans:** Thank you for raising this concern. However, **the observation that Llama-2-7B achieves better perplexity than Llama-3-8B on WikiText-2 is consistent with existing literature[1] and not indicative of an issue in our experimental setup.** One potential explanation is that a larger vocabulary (128k for Llama-3-8B versus 32k for Llama-2-7B) inherently increases prediction difficulty for each token, resulting in a naturally higher perplexity. It is also important to note that Llama-3-8B-Inst demonstrates improved zero-shot accuracy over Llama-2-7B, with average scores of 67.99 and 65.05, respectively. To ensure reproducibility, we have provided code implementation in revision. We hope this explanation addresses your concern.
>
> [1] SpinQuant: LLM quantization with learned rotations: https://arxiv.org/pdf/2405.16406
>
> > (Q2): Rather than compressing weights, since KV cache compression is desired, can a method such as ESPACE [1] be used to project activations to a low rank directly?
>
> **Ans:** Thank you for the suggestion. Since ESPACE was published on October 7, 2024, the same date as our submission deadline, we could not include a discussion or comparison in our initial submission.  Broadly, ESPACE can serve as an alternative approach for deriving low-rank projection matrices. Palu derives low-rank matrices $\mathbf{W}\approx\mathbf{A}\mathbf{B}$ through weight decomposition. ESPACE performs calibration on input x and then uses eigenvector decomposition to obtain the low-rank projection matrix $\mathbf{P}$. During inference, ESPACE operates as $\mathbf{h}= \mathbf{x}\mathbf{A'}$ and $\mathbf{y}=\mathbf{h}\mathbf{B'}$ with $\mathbf{A'}=\mathbf{P}$ and $\mathbf{B'}=\mathbf{W}\mathbf{P}$ pre-computed, eventually achieving the same down-projection and reconstruction style of forwarding pass as the weight decomposition style adopted in Palu.
>
> It would be interesting in future work to compare the accuracy differences by deriving a low-rank projection matrix via decomposing weight directly or other techniques. Furthermore, we note that our proposed GPU kernel optimizations and quantization-friendly enhancements—such as fusing the Hadamard Transform—are agnostic to the algorithm to derive a low-rank projection matrix. Thus, these techniques can provide a solid foundation for further research integrating ESPACE or other methods.
>
> [1] ESPACE: Dimensionality Reduction of Activations for Model Compression https://arxiv.org/abs/2410.05437

---

> > ### Comment · Reviewer_pJHR · 2024-11-20
> > **Response**
> >
> > Thank you for responding.
> >
> > I appreciate the discussion provided as response to my questions. However, with regards to the weaknesses I had raised in my review, the authors have only descriptive responses with no hard evidence to support their rebuttal.

---

> ### Author Response · Authors · 2024-11-20
>
> Thank you for your follow-up. We believe our previous responses, including the global response, have comprehensively addressed your concerns. For Weakness 1, we strongly encourage you to refer to the expanded discussion in the global response, which explains how Palu leverages the compute-memory trade-off to achieve speedups despite the additional computational requirements.
>
> For reviewer convenience, we include Table R3, extracted from Figure 6 of our paper, with extended coverage to test sequence lengths up to 256k. The results demonstrate that Palu consistently achieves significant speedups (1.51x ~ 1.95x) across various sequence lengths.
>
> **Table R3.** Latency evaluation of computing attention score. All time measurements are reported in microseconds.
> |  Seqlen          | 4k     | 8k     | 16k    | 32k    | 64k    | 128k    | 256k    |
> |-------------------|--------|--------|--------|--------|--------|----------|---------|
> | Baseline   | 54.37  | 103.42 | 200.70 | 346.96 | 634.66  | 1212.03 | 2365.38 |
> | Palu (gs-4)       | 35.90  | 55.33  | 102.92  | 199.68   | 401.87  | 794.62  | 1555.46 |
> | Speedup    | 1.51   | 1.87   | 1.95   | 1.74        | 1.58      | 1.53      | 1.52    |
>
> For Weakness 2, which pertains to the conceptual understanding of our paper, we note that a global response is also provided to clarify Palu’s distinction from related works focusing on parameter reduction. Regarding the remaining concerns related to the experimental setup and results (e.g., perplexity results), clarifications and references have also been provided.
>
> If there are specific areas or additional “concrete evidence” you feel are missing, we kindly request clarification so we can further support our rebuttal and address your concerns effectively.

---

> > ### Comment · Reviewer_pJHR · 2024-11-21
> > **Help me understand when a GEMM is faster than a GEMV**
> >
> > Rather than handpicking end-to-end numbers where one could question the design choice of the baseline, I think a strong rebuttal would be to compare the time spent in executing GEMVs of appropriate dimensions corresponding to the decoding phase to the proposed implementation which introduces extra GEMMs. I would like to see under which scenario is a GEMM faster than a GEMV. This depends on the dimensionalities of operators, so please conduct such study to convince me. Intuitively, a GEMM will always be slower than a GEMV simply because a GEMM is a stack of multiple GEMVs - but I am willing to change my position if you thoroughly show that there are corner cases where that assertion does not hold.
> >
> > Measuring GEMM and GEMV latency should be fairly simple. As an extra step, the authors are also encouraged to correlate their findings with NVIDIA's nsight systems profiles of their decoding implementation.
> >
> > If the above can be addressed, then it will be a start in better responding to the weaknesses I had raised. The answers to the questions (not weaknesses) were quite good - please add those to your revision.

---

> ### Author Response · Authors · 2024-11-23
> **Response to Reviewer pJHR (1/2)**
>
> We are glad to hear that our responses to the questions were well-received. Yes, we have included a discussion of ESPACE in Appendix K of the revised manuscript, as suggested. Additionally, thank you for providing further clarification regarding your concern about the additional overhead of GEMM required by Palu. Please refer to the subsequent comments for a detailed discussion.

---

> ### Author Response · Authors · 2024-11-23
> **Response to Reviewer pJHR (2/2)**
>
> # The Explanation Behind Palu's Speedup
>
> We thank the reviewer for your insightful comments. As the reviewer noted, GEMM involves additional FLOPs compared to GEMV, as it processes a stack of vector multiplications. The reviewer mentioned that the GEMM operation is needed in our proposed Palu framework. However, **we emphasize that Palu significantly reduces the global memory (DRAM) read/write traffic**, which typically dominates the latency of attention operations in LLM decoding and, therefore, can still achieve high speedups.
>
> **Why Palu Requires Fewer Global Memory Access.** In the baseline approach, the attention scores for each head are computed directly via $\mathbf{q}_i\mathbf{K}_i^T$, which is a GEMV with $\mathbf{q}_i\in\mathbb{R}^{1\times d_h}$ and $\mathbf{K}_i \in \mathbb{R}^{L\times d_h}$ denote the query and full-rank key matrix, respectfully. In Palu, we store the low-rank latent $\mathbf{H_i}\in \mathbb{R}^{L\times r}$ and the reconstruction matrix $\mathbf{B_i}\in \mathbb{R}^{r\times d_h}$ in place of the full rank key. The full-rank Key matrix $\mathbf{K}_i$ can then be reconstructed via $\mathbf{H}_i\mathbf{B}_i$.
>
> Instead of naively reconstructing $\mathbf{K_i}$ with a GEMM operation followed by a GEMV to compute attention scores, **the Palu framework employs tiling and operator fusion to streamline the process into a single optimized operation**. For a detailed description of our kernel designs and optimizations, please refer to Appendix B. This approach avoids materializing the full $\mathbf{K_i}$ in global memory by keeping intermediate activations (e.g., reconstructed $\mathbf{K_i}$) in the GPU’s shared memory and registers. Consequently, Palu significantly reduces global memory access, requiring only $\mathbf{q_i}$, $\mathbf{H_i}$, and $\mathbf{B}$ - which are substantially smaller compared to $\mathbf{q_i}$ and $\mathbf{K_i}$ in the baseline method.
>
>
>
> **Quantitive Analysis.** To provide a detailed comparison, we analyze the global memory traffic (reads/writes) and FLOPs required for attention score computation for the baseline GEMV approach and Palu, considering the different decomposition granularities proposed in our paper: M-LRD, G-LRD, and J-LRD. The quantitative results are summarized in Table R5. For this evaluation, we set a 75% compression rate for the key matrix, as the setup used in our paper, and a sequence length of 32k. The latency values average over 100 runs, with global memory traffic measurements taken using the NVIDIA Nsight Compute profiler.
>
> Table R5. Global memory (off-chip) footprint, FLOPs, and latency evaluation on calculating attention score.
> | **Method**              | **Mem W/R (MB)** | **FLOPs (GB)** | **Avg Latency (us)** | **Speedup** |
> | ----------------------- | ------------------ | -------------- | -------------------- | ----------- |
> | Baseline (Batched GEMV) | 272.83             | 0.27           | 290.61               | -           |
> | Palu (M-LRD)            | 68.07              | 8.86           | 76.91                | 3.78        |
> | Palu (G-LRD)            | 69.90              | 34.63          | 181.21               | 1.60        |
> | Palu (J-LRD)            | 78.84              | 275.146        | 1384.12              | 0.21        |
>
>
> From Table R5, it is evident that Palu significantly reduces global memory traffic, achieving up to a 75% reduction compared to the baseline GEMV approach. This reduction directly translates to latency improvements, as memory access is the primary bottleneck in attention computation. For example, in the M-LRD case, memory traffic decreases by nearly 4x, leading to proportional latency improvements of 3.78x. This demonstrates that even with additional computation, significant speedups can be achieved by optimizing memory access.
>
> However, as the reviewer noted, additional FLOPs could harm the latency. For the J-LRD case, where reconstruction FLOPs are highest, the latency increases by nearly 4.77x compared to the baseline. This reflects the trade-off between reconstruction cost and latency improvements. **The G-LRD variant balances accuracy and efficiency by grouping heads, achieving a 1.6x speedup while maintaining moderate computational overhead**.
>
> In conclusion, while Palu introduces additional computation from reconstruction, its significant memory traffic reduction, combined with the G-LRD design and optimized reconstruction kernel with tiling and operator fusion, makes it an effective solution for memory-bound LLM decoding. We appreciate the reviewer’s feedback, which helped us clarify these advantages. For further validation, we have uploaded a detailed report from the Nsight Compute System, as suggested.

---

> > ### Comment · Reviewer_pJHR · 2024-11-23
> > **Compute-bound vs memory-bound**
> >
> > I thank the authors for following up. I think one confusion the authors have is that they believe all computation is memory-bound, meaning that the latency is always dominated by the time it takes to fetch data from global memory. This is generally the case for GEMV-type operations where weights are fetched and used only once. GEMM-type operations are much different - when tiling and weight reuse come into the picture, the latency of computation becomes dominated by compute. Just FYI, this is why companies like NVIDIA built tensor cores. The authors can check on Nsight compute what is the tensor core utilization in the two cases to better understand this. The tool should also report memory bandwidth utilization. It will be very clear that the two are complementary.

---

> ### Author Response · Authors · 2024-11-23
> **Further Clarification Regarding Palu's Speedup**
>
> Thank you for your response and for raising this important distinction. We want to clarify that we do NOT assert all computations are inherently memory-bound. Factually, Table R5 from our previous response demonstrates that LLM decoding with low-rank compressed KV-Cache reconstruction can become compute-bound, potentially leading to slowdowns. Consequently, this observation motivated us to propose a novel G-LRD design to address this issue effectively.
>
> To further illustrate, we have expanded Table R5 into Table R6 with estimated data movement and compute times, calculated using the peak global memory bandwidth and compute throughput of the RTX 4090. We classify an operation as memory-bound if its latency is dominated by memory access time, and compute-bound if it is determined by computational throughput [1, 2]. These estimates highlight that GEMV operations in the baseline and M-LRD remain memory-bound. However, with Palu’s G-LRD and J-LRD, the workload shifts to being compute-bound. **Importantly, by leveraging modern GPUs’ high compute throughput and maintaining a balanced computational overhead, G-LRD achieves a 1.6x speedup over the baseline.**
>
> We hope this additional clarification addresses your concern, and we are happy to provide further explanations if needed.
>
>
> Table R6: Comparison of total memory traffic (W+R), FLOPs, estimated compute and memory times, and measured average latency for baseline and Palu's different decomposition methods.
>
> | **Method**        | **Total Mem (MB)** | **FLOPs (GB)** | **Est. Comp Time (us)** | **Est. Mem Time (us)** | **Avg. Latency (us)** | **Op Type** |
> | ----------------- | ------------------ | -------------- | ---------------------- | ----------------- | ---------------- | ------------- |
> | Baseline (GEMV) | 272.83             | 0.27           | 1.38                   | 248.08            | 290.61           | Mem Bound     |
> | Palu (M-LRD)      | 68.07              | 8.86           | 45.53                  | 62.31             | 76.91            | Mem Bound     |
> | Palu (G-LRD)      | 69.90              | 34.63          | 177.98                 | 63.05             | 181.21           | Comp Bound     |
> | Palu (J-LRD)      | 78.84              | 275.146        | 1414.20                | 69.82             | 1490.2           | Comp Bound     |
>
> [1] LLM Inference Unveiled: Survey and Roofline Model Insights https://arxiv.org/abs/2402.16363
>
> [2] Roofline: An Insightful Visual Performance Model
> for Floating-Point Programs and Multicore Architectures https://dl.acm.org/doi/10.1145/1498765.1498785

---

> > ### Comment · Reviewer_pJHR · 2024-11-24
> > **Raise to 5**
> >
> > I thank the authors for their persistent responses. I am still not convinced that introducing a GEMM-type operation within the decoding phase is a good idea. But the authors seem adamant, and while the answers and measurements are not too convincing, there may be some value in the proposal. I will raise the score to 5 and let AC be the final judge on that.

---

> > > ### Author Response · Authors · 2024-12-02
> > > **Follow-up on Responses to Reviewer pJHR**
> > >
> > > Dear Reviewer pJHR,
> > >
> > > With the discussion period concluding on Tuesday, we wanted to follow up regarding your concerns about introducing a GEMM-type operation during the decoding phase. We have carefully considered your comments and addressed them to the best of our ability in our previous response and supporting experiments.
> > >
> > > If there are specific aspects of our clarifications or experiments that you still find unclear, we would appreciate it if you could provide additional details so that we can address them before the discussion period ends.
> > >
> > > Thank you once again for your time and effort in reviewing our work.
> > >
> > > Sincerely,
> > >
> > > Palu Authors

---

> > > > ### Comment · Reviewer_pJHR · 2024-12-02
> > > > **Summary of our discussion**
> > > >
> > > > As this is the last day for reviewers to respond, let me summarize where we stand. I had initially provided a score of 3, having found several issues with the paper. The most critical aspect was the fact that the proposal introduces more compute during the decoding phase. There has been a lot of discussion, and I thank the authors for those. Of course, we can cherry-pick scenarios where the proposal makes sense and this has been shown in the responses. As such, I had raised my score to 5 - and this is where I stand. I am not convinced about the proposal, but was willing to recommend a borderline reject in order to give the benefit of the doubt - so that should the Area Chair decide to supersede this recommendation - I would not object.

---

> ### Author Response · Authors · 2024-11-24
> **Follow-up on Addressing the Concern of Additional GEMM for Reconstruction.**
>
> Thank you again for your continued engagement and for raising your score. We sincerely appreciate the time and effort you have dedicated to reviewing our work.
>
> We believe that the experimental results presented in Table R6, along with the discussion on the compute-bound and memory-bound nature of the operations, directly address the previously raised concerns on the additional GEMM-type operation. Also, we proved that our approach can still practically bring speedups.
>
> If there are specific aspects of our results or arguments that you find unconvincing, we would greatly appreciate your clarification. Gaining a deeper understanding of your concerns will allow us to address them more effectively and refine our presentation for future paper readers. We also welcome any further questions or clarifications that may help address other parts of the work where you feel additional discussion is needed.

---

> ### Author Response · Authors · 2024-11-24
> **Further Clarification Regarding the Main Concern on Additional GEMM Requirements for Reconstruction**
>
> > (W1): In the Palu framework, a full GEMM is required to rematerialize each key and value matrix to be used in the attention module. This adds a lot of complexity, particularly when decoding a large output sequence length.
>
> We would like to further clarify a misunderstanding from this comment. In Palu, there is no need to rematerialize the full Value matrix, whether for RoPE-based or non-RoPE attention. This is because the reconstruction matrix for the Value can be fused with the next linear layer's weight (see Sec 3.1.1, Lines 185–201), enabling a highly efficient inference process for the Value. Therefore, the reconstruction process described in the previous comment is required only for the RoPE-based Key Cache.
>
> We hope that this further clarification, together with the demonstrated practical speedups discussed previously, could further address your concern regarding the additional GEMM requirements for performing reconstruction in the proposed Palu framework.

---

> ### Author Response · Authors · 2024-12-03
> **Summary of Discussion of Additional GEMM Requirements and Palu's Efficiency**
>
> We sincerely thank the reviewers for their time and effort throughout the discussion session.
>
> As the discussion cycle concludes today, we want to summarize the experiments and corresponding discussions regarding the introduction of additional GEMM operations in Palu:
>
> **1. Comprehensive latency measurement of attention score calculation with reconstruction on various sequence lengths.** On benchmarking attention score computed with additional GEMM required for reconstruction, our evaluations comprehensively evaluate the sequence length from 4k to 256k (**Table R3**). Also, we set the tensor configurations (e.g., hidden dimensions, number of attention heads) identical to the parameters of Llama-2 to ensure benchmarking is aligned with real-world LLM usage.
>
>
> **2. Quantitative Analysis.** The detailed analysis in **Table R6** provides insights into memory footprint reduction, validated using the Nsight Compute profiler, and additional FLOPs quantified for each decomposition scenario (e.g., M-LRD, G-LRD, J-LRD). These metrics are coupled with explanations of how speedups are achieved by ensuring that the additional FLOPs remain manageable within GPU compute capabilities.
>
> **3. End-to-End Benchmark.** Beyond the micro-benchmarking of attention score computation, our paper also incorporates broader end-to-end decoding latency benchmarks. These benchmarks highlight Palu’s overall decoding efficiency improvements in practical usage scenarios (**Sec 4.5 and Figure 5**).
>
> By including a broad spectrum of sequence lengths, decomposition methods, and practical use cases, we ensure that the findings are **not cherry-picked** at all but instead **comprehensively** demonstrate Palu's effectiveness in improving LLM decoding efficiency.

---

### Official Review · Reviewer_2i3G · 2024-10-31

**Soundness:** 4
**Presentation:** 4
**Contribution:** 3
**Rating:** 6
**Confidence:** 5

**Summary:**

The paper introduces Palu, a KV-Cache compression framework that reduces memory usage in LLM inference by applying low-rank projection to the hidden dimension. Palu decomposes linear layers into low-rank matrices, caching compressed intermediate states and reconstructing KV values on the fly. Key features include medium-grained low-rank decomposition, automated rank search, quantization compatibility enhancements, and a GPU-optimized matrix fusion kernel. Experiments show that Palu compresses KV-Cache by 50% and boosts attention module speed by up to 1.87×.

**Strengths:**

S1: The method creatively combines SVD algorithms with KV-cache compression, optimizing speed and grouping different attention heads with varied compression rates, which is insightful.

S2: The experiments are comprehensive, measuring both the algorithm’s accuracy and speed.

S3: The figures and tables are clear, and the descriptions are precise.

**Weaknesses:**

W1: In Section 3.1, the statement "However, this approach poses significant computational challenges during runtime that make it impractical for deployments" lacks experimental results and theoretical support, even though this claim is foundational to the proposed method.

W2: There is no theoretical proof for the accuracy of the proposed algorithm, such as bounded error analysis.

W3: The time complexity of the algorithm is not analyzed.

W4: In Section 4, the relationship between "compression rate" and "Avg. Bits" is unclear. Additionally, definitions of metrics such as "compression rate" are not specified.

W5: The code is not open-sourced, making reproducibility uncertain.

**Questions:**

Q1: Corresponding to W1, provide theoretical or empirical evidence for the statement in Section 3.1 that " However, this approach poses significant computational challenges during runtime that make it impractical for deployments "

Q2: Corresponding to W2, add a theoretical derivation of the algorithm's error bounds.

Q3: Corresponding to W3, include an analysis of the algorithm's time complexity.

Q4: Corresponding to W4, clarify the relationship between "compression rate" and "Avg. Bits" in Section 4, along with definitions of related metrics.

Q5: It is recommended to open-source the code.

**Details Of Ethics Concerns:**

I believe this paper does not require an ethics review.

---

> ### Author Response · Authors · 2024-11-19
> **Response to Reviewer 2i3G**
>
> > (Q1). The statement "However, this approach poses significant computational challenges during runtime that make it impractical for deployments" lacks experimental results and theoretical support
>
> **Ans:** Thanks for the good suggestions. To provide additional clarity, we have benchmarked the time required for performing SVD directly on the KV-Cache in comparison to the time needed for a forward pass through a decoder block, as shown in the Table R2 below:
>
> **Table R2.** Latency comparison between performing SVD on the KV-Cache and a single forward pass through a transformer decoder block, using the model configuration of Llama-2-7B.
>
> | Seqlen                   | 32k    | 64k   | 128k   |
> | ------------------------ | ------ | ----- | ------ |
> | Decoder Block    | 0.231s | 0.65s | 1.93s  |
> | SVD                  | 9.24s  | 9.90s | 11.44s |
>
> These results highlight that performing SVD on the fly introduces significant computational overhead. Specifically, runtime SVD is approximately 5–10 times slower than a forward pass through the decoder block, making it infeasible for real-time inference.
>
> Additionally, SVD is highly precision-sensitive and requires higher precision (e.g., fp32 or fp64), introducing further type casting overhead if the base model is executed in low precision (e.g., fp16). This sensitivity to precision adds to the computational burden.
>
> Given these findings, Palu adopts an alternative approach by performing weight matrix decomposition offline to yield low-rank projection matrices. With these decomposed low-rank matrices, Palu achieves KV-Cache compression by caching the intermediate latent representations, effectively bypassing the runtime overhead associated with online SVD.
>
> > (Q2, Q3). Lack of theoretical proof, such as bounded error analysis and analyzation.
>
> **Ans:** Thank you for raising this point. We acknowledge that providing a formal theoretical proof, such as bounded error analysis would strengthen the paper. While we lack theoretical analysis, we have conducted extensive experiments, as you have recognized, to validate the robustness, accuracy and efficiency of Palu under various conditions.
>
> > (Q4).  In Section 4, the relationship between "compression rate" and "Avg. Bits" is unclear. Additionally, definitions of metrics such as "compression rate" are not specified.
>
> **Ans:** Thank you for the suggestion. To clarify, the term “compression rate” refers to the overall reduction in KV-Cache size achieved by applying low-rank compression alone or in combination with quantization compared to the uncompressed baseline. “Avg. Bits” indicates the average number of bits required to store each element in the KV-Cache, considering quantization parameters, such as shared scales, that apply to a group of elements. We have updated Section 4 to be more clear.
>
> > (Q5). The code is not open-sourced
>
> **Ans:**  Thanks for your recommendation. We have uploaded the code via the OpenReview platform for your review. Please refer to the supplementary materials for further details.

---

> > ### Comment · Reviewer_2i3G · 2024-11-24
> > **Response to the Rebuttal.**
> >
> > I believe that the issues I raised regarding Q1, Q4, and Q5 have been well addressed. Q1 has been validated with relevant experiments demonstrating the correctness of the statements, Q4 has been appropriately revised, and for Q5, the code has been open-sourced. However, for Q2 and Q3, it would be better to provide theoretical derivations for the algorithm's error bounds and its time complexity.

---

> ### Author Response · Authors · 2024-11-23
> **Message to Reviewer 2i3G**
>
> Dear Reviewer 2i3G,
>
> We sincerely appreciate your time and support for our work in the initial review—it truly means a great deal to us!
>
> As a gentle reminder, the rebuttal cycle will conclude next Tuesday. We would be grateful if you could take a moment to review the additional analysis and clarifications we have provided in response to your comments. Additionally, the open-sourced code has now been uploaded. We would also like to confirm whether these updates have adequately addressed your concerns.
>
> If you have any further feedback or suggestions, please don’t hesitate to let us know. Thank you again for your valuable time and effort!
>
> Best regards,
>
> Authors

---

> ### Author Response · Authors · 2024-11-26
> **Follow-up Response to Reviewer 2i3G (Add Error bounds and Complexity Analysis)**
>
> Thank you very much for your thoughtful and constructive feedback on our work. We greatly appreciate the time and effort you have taken to provide your insights.
>
> We want to follow up regarding your comments on Q2 and Q3. To address your concerns, we have made the following additions to our revised manuscript:
>
> 1. In Appendix L, we have included proof to demonstrate that the error on the KV-Cache induced by performing low-rank decomposition can be bounded.
> 2. In Appendix M, we have provided a detailed derivation and summary of the computing and memory complexity required to compute the attention score when applying the proposed Palu.
>
> We hope this additional information can help address your concerns, and we are grateful for your valuable suggestions that have helped improve the rigor and clarity of our work.
>
> Best regards,
>
> Palu Authors

---

> > ### Author Response · Authors · 2024-11-27
> > **Thank for Your Insights and Follow-Up on the Responses to Reviewer 2i3G**
> >
> > Dear Reviewer 2i3G,
> >
> > We sincerely appreciate the time and effort you have devoted to reviewing our work and providing thoughtful feedback. Your insights have been invaluable in helping us refine and enhance the quality of our manuscript.
> >
> > As the deadline for revising the PDF approaches today, we wanted to follow up to ensure that we have addressed all of your concerns thoroughly and have further improved the manuscript based on your suggestions. If there are any additional questions or aspects that require clarification, we would be more than happy to provide further details. If you feel that our revisions and responses have adequately addressed your feedback, we would greatly appreciate it if this could be reflected in your score.
> >
> > Thank you once again for your thoughtful review and for sharing your expertise. We deeply value your constructive feedback.
> >
> > Sincerely,
> >
> > Palu Authors

---

> ### Comment · Reviewer_2i3G · 2024-11-28
> **Follow-up Response to the Rebuttal.**
>
> I sincerely thank the authors for their persistent responses.  I will raise AC be the final judge on the responses.

---

> ### Author Response · Authors · 2024-12-02
>
> Dear Reviewer 2i3G,
>
> Thank you for your reply and detailed review of our paper. We truly appreciate your positive feedback and the rating. We also welcome any further questions or concerns you may have during the remaining two days of the discussion period.

---

### Official Review · Reviewer_Fsn9 · 2024-11-02

**Soundness:** 2
**Presentation:** 2
**Contribution:** 3
**Rating:** 6
**Confidence:** 4

**Summary:**

The paper introduces Palu, a novel framework designed to compress the KV-cache of large language models (LLMs) using low-rank projection. Unlike existing techniques, such as quantization or token eviction, which may overlook redundancies in hidden dimensions, Palu effectively captures these latent efficiencies. Notably, it provides a post-training compression solution, in contrast to low-rank methods like GQA and MLA that require training from scratch. This approach allows Palu to strike a balance between memory efficiency, accuracy, and inference speed.

**Strengths:**

1. **Post-Training Solution:** Unlike MLA, which requires retraining from scratch, Palu can be applied post-training. This versatility allows for the compression of existing models without additional, costly training cycles.
2. **Balanced Accuracy and Efficiency:** The group-head low-rank decomposition (G-LRD) balances accuracy and computational efficiency. It preserves essential information while reducing reconstruction overhead compared to per-head and joint-head decompositions.
3. **Automated Rank Allocation:** By using Fisher information to dynamically assign ranks to matrices based on their importance, Palu ensures compression without significant accuracy loss. This approach optimizes each layer's sensitivity to compression.
4. **Quantization-Friendly Design:** Palu addresses quantization-induced outliers with a Hadamard transformation, effectively supporting low-bit quantization. This further enhances compression without compromising accuracy, yielding competitive perplexity scores compared to quantization-only methods.

**Weaknesses:**

1. **Inefficiency with Positional Encoding**: Palu requires reconstructing the original key cache when calculating positional encodings, which may reduce computational efficiency.
2. **Performance on Complex Tasks**: Palu-50% does not outperform lower-compression 2-bit quantized KV-Cache methods, particularly on complex tasks like Code and Synthetic. This limitation suggests that Palu’s effectiveness may decrease in more challenging scenarios, where high compression ratios can impact accuracy.

**Questions:**

Is it possible to compress position embedding by similar way to remove the key cache reconstruction in attention computing? Or do some better tricks in attention computing rather than reconstructing key states?

---

> ### Author Response · Authors · 2024-11-19
> **Response to Reviewer Fsn9**
>
> > (W1) Inefficiency with Positional Encoding.
>
> **Ans:** While reconstruction might initially appear to introduce overhead, our proposed techniques—such as G-LRD and a co-designed GPU kernel—enable Palu to achieve a notable 1.89x speedup over standard attention by effectively balancing memory footprint and compute. For a more detailed explanation, please refer to the response in the global rebuttal.
>
> > (W2) Performance on Complex Tasks.
>
> **Ans:**  We acknowledge that Palu-50% does not outperform lower-compression 2-bit quantized KV-Cache methods on complex tasks such as Code and Synthetic. However, quantization and hidden-dimension reduction are fundamentally distinct techniques with different strengths. Rather than comparing them head-to-head, we believe their complementarity is more important.
>
> To illustrate this, we incorporated quantization-friendly enhancements into Palu’s low-rank compression. For instance, on coding tasks, Palu-30% combined with simple per-token quantization achieves comparable accuracy while delivering an additional 30% KV-Cache reduction (as shown in Table 3). Moreover, Appendix E demonstrates that Palu is compatible with Parameter Efficient Fine-Tuning (PEFT) methods like LoRA, which can further recover accuracy. These results underscore Palu’s flexibility and effectiveness in combining with other approaches to tackle challenging scenarios.

---

> ### Author Response · Authors · 2024-11-23
> **Message to Reviewer Fsn9**
>
> Dear Reviewer Fsn9,
>
> We sincerely appreciate your time and thoughtful feedback on our work! We are glad to see that you lean toward acceptance of our submission—it truly means a great deal to us.
>
> As a gentle reminder, the rebuttal cycle will conclude next Tuesday. We would be grateful if you could take a moment to review the additional experimental results and detailed explanations we have provided, as well as the global response. We also want to confirm whether our responses have adequately addressed your concerns.
>
> Please feel free to let us know if you have any additional feedback or points of clarification—we would be happy to address them. Thank you again for your valuable time and effort.
>
> Best regards,
>
> Authors

---

> > ### Comment · Reviewer_Fsn9 · 2024-11-26
> > **Response to Rebuttal**
> >
> > I have reviewed the rebuttal and the revised version with the updated appendix, which addresses my question about Palu's performance and responds to other reviewers' concerns. I believe Palu represents valuable work, and based on these updates, I have decided to raise my rating from 5 to 6.
> >
> >
> > I do, however, have a follow-up question regarding W2. In Table 1's perplexity test, Palu exhibited a greater performance drop on Llama3-8B (with 8 KV heads) compared to Llama2-7B (with 32 KV heads). This observation raises concerns about Palu's scalability and performance on newer LLMs, such as Qwen2.5, which employ significantly compressed KV head states (e.g., Qwen2.5-1.5B with only 2 KV heads and Qwen2.5-7B with 4 KV heads).
> >
> >
> > While I understand the time limits, I hope that in future updates, Palu can include additional testing with models different on Q/KV head ratios to better demonstrate its robustness across different architectures.

---

> ### Author Response · Authors · 2024-11-27
> **Follow-up Response to Reviewer Fsn9**
>
> Thank you for your thoughtful review and for taking the time to evaluate our rebuttal and updated paper. We are delighted to hear that you find our work valuable and appreciate your decision to raise the score. Below, we provide responses to your follow-up question:
>
> > Palu's performance on newer LLMs, such as Qwen2.5, which employ significantly compressed KV head states.
>
> We appreciate your insightful suggestions and for raising concerns regarding Palu’s performance on LLM architectures with varying KV-head configurations. Table R7 provides experimental results on Qwen2.5-7B (4 KV heads) and Qwen2.5-1.5B (2 KV heads), showcasing Palu’s adaptability to architectures with compact KV-Caches.
>
> **Table R7.** Zero-shot accuracy of Palu at different compression rates on Qwen2.5 models
> | Model        | Method   | OBQA | Hella | PIQA | ARC-e | ARC-c | Wino | Avg. Acc |
> | ------------ | -------- | --------------------- | -------------------- | ---------- | -------------- | ------------------------ | ---------------- | ------- |
> | Qwen2.5-7B   | Baseline | 47.20%                | 78.93%               | 78.78%     | 80.39%         | 51.02%                   | 72.93%           | 68.21%  |
> |              | Palu-30% | 45.60%                | 77.80%               | 77.75%     | 81.06%         | 53.75%                   | 70.40%           | 67.73%  |
> |              | Palu-50% | 43.80%                | 74.52%               | 77.37%     | 76.85%         | 52.30%                   | 71.27%           | 66.02%  |
> |              |          |                       |                      |            |                |                          |                  |         |
> | Qwen2.5-1.5B | Baseline | 40.80%                | 67.72%               | 75.79%     | 75.51%         | 45.14%                   | 63.38%           | 61.39%  |
> |              | Palu-30% | 39.60%                | 64.88%               | 73.43%     | 72.93%         | 42.75%                   | 62.47%           | 59.42%  |
> |              | Palu-50% | 35.20%                | 58.43%               | 69.70%     | 62.21%         | 35.07%                   | 60.85%           | 53.58%  |
>
>
> Seeing from Table R7, **Palu demonstrates strong performance on Qwen2.5-7B, a model variant with optimized KV states featuring only 4 KV-heads.** Despite its compact KV-Cache design, which inherently makes compression more challenging, Palu achieves near-lossless accuracy at a 30% compression rate, incurring only a 0.48% average drop across six datasets. **Even at a more aggressive 50% compression rate, Palu maintains robustness, with only a 2.2% average accuracy drop.** These results confirm Palu’s adaptability and effectiveness in compressing models with fewer KV-heads, demonstrating its capability to handle compact KV-Caches while preserving high performance across tasks.
>
> For Qwen2.5-1.5B, which possesses only 2 KV-heads, we observe that fully preserving performance with Palu becomes more challenging, particularly at an aggressive 50% compression rate. This phenomenon can be attributed to fewer KV-heads, which naturally results in smaller KV-Caches with higher information density, leaving less redundancy for compression techniques to exploit. However, our experiment shows that Palu still **achieves robust accuracy at a competitive 30% compression rate, preserving a significant portion of the baseline performance across tasks with only 1.97% accuracy degradation on average.** This highlights Palu’s ability to handle demanding scenarios while still delivering meaningful compression benefits, even for architectures with ultra-compact KV-Caches.
>
> All in all, these findings demonstrate Palu’s scalability and robustness, effectively showing that it can identify and exploit redundancy even in the model architecture that KV-Cache have already been highly optimized.
>
> We hope these additional experiments address your concerns regarding the robustness of the proposed Palu framework.

---

> > ### Author Response · Authors · 2024-12-02
> > **Looking Forward to Reviewer Fsn9's Feedback**
> >
> > Dear Reviewer Fsn9,
> >
> > As we approach the end of the discussion period, we are following up on our previous responses to your questions. We would greatly appreciate it if you could take a moment to review our responses on addressing the robustness concern and the additional results on Qwen2.5 that we have provided.
> >
> > We also welcome any further questions or concerns you may have during the remaining two days of the discussion period. If our responses and new results have sufficiently addressed your concerns, we would be grateful for your further feedback or an updated assessment.
> >
> > Thank you once again for your time, thoughtful feedback, and the effort you have dedicated to reviewing our work.
> >
> > Sincerely,
> >
> > Palu Authors

---

> > > ### Comment · Reviewer_Fsn9 · 2024-12-02
> > > **Response to Palu Authors**
> > >
> > > Thanks for your additional exploration. Based on the results, Palu demonstrates strong robustness at 30% compression, even with a very small KV cache hidden size. An additional question: is there no difference between joint-head and grouped-head approaches in 2 KV-head models?

---

> ### Author Response · Authors · 2024-12-02
> **Response to  Reviewer Fsn9**
>
> Thanks for your response. Below, we respond to your follow-up question:
>
> > Question: Is there no difference between joint-head and grouped-head approaches in 2 KV-head models?
>
> **Ans:** Yes, you are correct. In the specific case of a 2-head model, the grouped-head approach is equivalent to the joint-head approach. This equivalence arises because grouping in G-LRD encompasses all available heads into a single group ($n_g = 1$).
>
> As the reconstruction overhead for keys scales with the number of heads, the cost is significantly reduced in scenarios with extremely few heads. Consequently, Palu can leverage this reduced memory footprint by low-rank compression to alleviate memory bandwidth constraints and translate it into latency improvements in memory-bound large language model (LLM) decoding.

---

### Official Review · Reviewer_12bo · 2024-11-04

**Soundness:** 3
**Presentation:** 3
**Contribution:** 3
**Rating:** 6
**Confidence:** 5

**Summary:**

This paper addresses the issue of high GPU memory consumption in KV Cache during LLM inference by proposing a novel approach that differs from traditional KV Cache compression methods (such as quantization and token-level compression). Specifically, it employs Singular Value Decomposition (SVD) to reduce the rank of the KV Cache matrix, thereby decreasing its memory usage. Building on this method, the paper further explores several enhancements, including: 1) merging multiple heads into a group for SVD to minimize precision loss; 2) adaptively adjusting the compression ratio based on the redundancy of information at different layers; and 3) integrating the Walsh-Hadamard transform to facilitate quantization of the compressed results.

**Strengths:**

1.	The approach proposed in this paper, which utilizes Fisher information to allocate compression ratios for each linear layer, is both effective and scalable. It offers valuable guidance for other KV Cache and model parameter compression efforts.
2.	This paper implements a CUDA kernel to maximize throughput efficiency.
3.	This paper proposes and experimentally validates the compatibility of its optimization methods with traditional techniques such as quantization and LoRA fine-tuning, ensuring their practical applicability in real-world environments.
4.	This paper conducts comprehensive ablation experiments to validate the effectiveness of each optimization module individually.

**Weaknesses:**

1.	The method proposed in this paper, which utilizes SVD for rank reduction of parameter matrices, is not novel, as it has been employed in various studies (e.g., Sharma et al., "The Truth is in There: Improving Reasoning in Language Models with Layer-Selective Rank Reduction," 2023). The corresponding optimization of KV Cache storage is consequently straightforward given the rank reduction of the parameter matrices.
2.	The paper does not provide open-source code, which raises concerns about the authenticity and reproducibility of their experimental results.
3.	The paper compares a limited number of related works; aside from the baseline, it only includes quantization-based methods such as Atom and KVQuant. However, token-level compression methods are also a key focus in the KV Cache compression field, which this paper completely overlooks. The authors should at least conduct comparative experiments and method combination studies with one of the state-of-the-art approaches in this area, such as H2O (Zhang et al., "H2O: Heavy-Hitter Oracle for Efficient Generative Inference of Large Language Models," 2024).

**Questions:**

1.	Suggestion: The authors might consider adding a mathematical derivation section to demonstrate that the SVD method has an upper bound on the accuracy error resulting from the rank reduction of parameter matrices. This would enhance the generalizability and applicability of the method across different models and datasets.

---

> ### Author Response · Authors · 2024-11-19
> **Response to Reviewer 12bo**
>
> > (W1) SVD for rank reduction of parameter matrices, is not novel, as it has been employed in various studies (e.g., LASER[1]). Consequently, the corresponding optimization of KV cache storage is straightforward.
>
> **Ans:** Thank you for highlighting the role of SVD in prior work for parameter matrix reduction. SVD has been utilized in studies like LASER for compressing parameter matrices; however, LASER did not consider the fundamental shift required to optimize KV-Cache storage through rank reduction on parameter matrices. Palu introduces a redefined caching strategy by replacing full-dimensional KV tensors with low-rank latent representations. Beyond this innovation, Palu contributes new decomposition schemes, custom GPU kernels, and quantization-friendly enhancements—none of which are implemented or required in LASER. The global response above provides a detailed explanation of Palu’s key contributions and the challenges we addressed to deliver an efficient and practical solution.
>
> [1] The Truth is in There: Improving Reasoning in Language Models with Layer-Selective Rank Reduction. https://arxiv.org/abs/2312.13558
>
> > (W2) The paper does not provide open-source code
>
> **Ans:** Thank you for raising this concern. We will disclose our open-source implementation upon acceptance of the paper. In the meantime, we have uploaded the code via the OpenReview platform for your review. Please refer to the supplementary materials for further details.
>
> > (W3) Lack of discussion with Token-level compression methods
>
> **Ans:** Thank you for bringing up this concern. We would like to first clarify that Palu focuses on hidden-dimension compression, which operates in a fundamentally orthogonal direction to token-level methods. Token-level compression techniques, such as H2O, optimize KV-Cache efficiency by identifying and discarding less significant tokens. In contrast, Palu compresses the hidden dimensions across all tokens in a sequence using a layer-specific compression rate. As such, these methods are not directly comparable but could potentially be integrated to further optimize KV-Cache efficiency.
>
> To demonstrate the complementarity of our approach, we conducted additional experiments by integrating Palu with SnapKV [1], a more recent and advanced token-sparsity method than H2O. The results can be found in Table 1. As Table 1 shows, Palu-30% combined with SnapKV achieves performance comparable to applying each method independently in 5 out of 6 subjects (14 out of 16 tasks), with only a minor accuracy drop (~6.85%) on Synthetic datasets. These results provide strong evidence that token-sparsity methods and hidden-dimension reduction techniques can effectively complement each other.
>
> **Table R1.** LongBench evaluation for Mistral-7B-v0.2 using the Token-Level Method, Palu, and the Combination of Both.
> | Method                  | Multi-QA (\#tasks=3) | Single-QA (\#tasks=3) | Summarization (\#tasks=3) | Few-Shot (\#tasks=3) | Code (\#tasks=2) | Synthetic (\#tasks=2) |
> | ----------------------- | -------------------- | --------------------- | ------------------------- | -------------------- | ---------------- | --------------------- |
> | Baseline                | 29.63                | 36.43                 | 28.10                     | 66.71                | 54.16            | 44.87                 |
> | SnapKV-75%          | 29.74                | 36.63                 | 27.97                     | 66.70                | 53.94            | 44.85                 |
> | Palu-30%             | 29.83                | 36.52                 | 27.48                     | 65.70                | 55.16            | 37.92                 |
> | Palu-30% + SnapKV-75% | 30.14                | 36.54                 | 27.51                     | 66.00                | 53.70            | 37.99                 |
>
> Additionally, we have expanded our discussion to cover the token-level compression methods, including H2O and SnapKV, in Appendix J of the revised submission. This provides a broader perspective on related work and situates Palu within the context of KV-Cache optimization research. We hope this helps to address your concern and clarifies the distinct focus and contributions of our approach.
>
> Note: We acknowledge the accuracy drop on Synthetic datasets and believe it could be mitigated by adopting strategies inspired by token-level methods, such as preserving important tokens during compression. However, such advanced enhancements extend beyond the scope of this paper, which focuses specifically on the compression of the hidden dimension. We consider exploring these synergies as our future work.
>
> > (S1) Consider adding a mathematical derivation section.
>
> **Ans:**  We acknowledge that providing theoretical proof, such as bounded error analysis, would further strengthen the paper. Thanks for your suggestion. We will consider adding a mathematical derivation section to future work.

---

> ### Author Response · Authors · 2024-11-23
> **Message to Reviewer 12bo**
>
> Dear Reviewer 12bo,
>
> We sincerely appreciate your time and constructive feedback on our work!
>
> We would like to kindly remind you that the rebuttal cycle will conclude next Tuesday. It would mean a great deal to us if you could take a moment to review the updated experimental results and detailed explanations we have provided, particularly addressing your concerns regarding the unique focus of Palu compared to related work and the associated discussion.
>
> Your support is vital to our work, and we would be immensely grateful for your consideration and feedback.
>
> Thank you again for your valuable time and effort.
>
> Best regards,
>
> Authors

---

> ### Author Response · Authors · 2024-11-27
> **Follow-up Response to Reviewer 12bo**
>
> Dear Reviewer 12bo,
>
> We would like to kindly follow up regarding our recent revision. In line with your suggestion, we have included a theoretical proof in Appendix L demonstrating that the error on the KV-Cache induced by the rank reduction of parameter matrices can be bounded.
>
> Once again, we sincerely thank you for your thoughtful feedback and the time you have dedicated to improving our submission.
>
> Sincerely,
>
> Palu Authors

---

> ### Author Response · Authors · 2024-12-02
> **Follow-up on Responses to Reviewer 12bo Before Discussion Deadline**
>
> Dear Reviewer 12bo,
>
> We hope we have addressed all your concerns. As the discussion period concludes on Tuesday, December 2, we wanted to follow up to see if you have any additional questions regarding our previous responses, clarifications, or supporting experiments. If there are specific aspects you feel require further explanation, we would greatly appreciate it if you could let us know so that we can address them promptly.
>
> We sincerely welcome any further questions or feedback you may have as we approach the end of the discussion period.
>
> Thank you once again for your time, thoughtful feedback, and the effort you have dedicated to reviewing our work.
>
> Sincerely,
>
> Palu Authors

---

### Author Response · Authors · 2024-11-19
**Global Rebuttal**

## **1. Discussion on Palu’s Computation Overhead and Design Rationale**

Given that multiple reviewers raised concerns regarding Palu’s computational cost, particularly for RoPE-based attention, we provide a comprehensive explanation to address these comments and clarify the design rationale behind Palu.

**Memory-Bound Nature of LLM Decoding.** The decoding process in large language models (LLMs) is predominantly memory-bound [4]. A key optimization target in this paper—the attention mechanism—requires frequent retrieval of large key-value matrices and sequential computations. These operations require continuous data movement between global memory and compute units, which places significant pressure on memory bandwidth. This characteristic, combined with the relatively low arithmetic intensity of these operations, results in the underutilization of computational resources. As illustrated by the Roofline Model, performance in memory-bound workloads is **constrained primarily by memory traffic rather than computational throughput.** Also, the powerful **compute units are often underutilized because the data required for computations cannot quickly be supplied from off-chip memory.**

**Trading Compute for Memory Pressure Alleviation** Given memory-bound characteristics, a compression technique that introduces additional computation can be beneficial and effectively accelerate the decoding process by reducing memory bandwidth usage. A notable example is KV-Cache quantization techniques, which introduce additional computation (e.g., dequantization) for memory footprint reduction. In Palu, we build on similar insight from the Roofline model perspective to optimize decoding. By balancing additional computation with memory bandwidth savings, Palu reduces memory requirements while accelerating performance. As demonstrated in **Section 4.5 and Table R4**, **Palu achieves up to a 1.89x speedup over uncompressed baselines in RoPE-based attention.** Furthermore, **Palu’s quantization-friendly design enables up to 2.91x speedups and a 7.59x reduction in memory usage when combined with quantization.**

**Table R4**. Normalized Speedup for RoPE-based attention module. Speedup numbers are extracted from Figure 5 in our paper.
| Method|4K|8K|16K|32K|64K|
|-|-|-|-|-|-|
|KIVI-4-bit|0.80|1.18|1.49|1.68|1.89|
|Palu-50%|1.04|1.24|1.58 |1.76|1.89|
|Palu-50%-4-bit|1.87 | 2.42 | 2.56| 2.75| 2.91|

**Managing Computational Overhead for Reconstruction.** As the Reviewer pjHR notes, it is crucial to ensure the introduced computation is within the manageable level. To achieve this, we designed a novel medium-grained low-rank decomposition (G-LRD) that reduces FLOPs for reconstruction within a manageable level while preserving accuracy. With G-LRD, Palu could successfully deliver acceleration, as seen in Table R5 and [discussion thread with reviewer pJHR](https://openreview.net/forum?id=LWMS4pk2vK&noteId=8Zx8nJnUir).

Together, Palu effectively balances memory-compute trade-offs, achieving significant speedups and memory reductions in memory-bound LLM decoding.

## **2. Difference between Palu and Prior SVD’s Literature and Design Challenge.**

As pointed out by reviewers, we clarify the fundamental distinction between Palu and weight decomposition works [1, 2, 3]. In the existing literature, SVD is commonly used to decompose parameter matrices into two low-rank matrix pairs. However, these methods focus on reducing parameters and FLOPs and do not explore opportunities to change the caching mechanism during LLM decoding fundamentally.

Palu leverages this overlooked opportunity by designing a compression framework for caching the low-rank latent representations. While this concept might appear conceptually straightforward, making it practically efficient is non-trivial, especially when dealing with RoPE-based attention. One of the critical challenges we address is minimizing the overhead associated with reconstructing the Key-Cache from the low-rank representations using the second decomposed matrix during inference. This led us to develop the group-head low-rank decomposition (G-LRD), which optimizes the decomposition granularity across attention heads. Unlike naive approaches like joint-head low-rank decomposition (J-LRD), which incur higher reconstruction costs and lead to non-negligible slowdowns, our proposed G-LRD achieves a 1.95× speedup at sequence length 16k (see Figure 6.), enabling efficient and practical deployment.

---
[1] The Truth is in There: Improving Reasoning in Language Models with Layer-Selective Rank Reduction. https://arxiv.org/abs/2312.13558

[2] Language model compression with weighted low-rank factorization. https://arxiv.org/abs/2207.00112

[3] Data-aware low-rank compression for large NLP models. https://openreview.net/forum?id=_sSHg203jSu

[4] LLM Inference Unveiled: Survey and Roofline Model Insights https://arxiv.org/abs/2402.16363

---

### Author Response · Authors · 2024-11-19
**General Response**

We sincerely thank all the reviewers for their thoughtful feedback and for taking the time to read our paper. We have uploaded a revised manuscript with polished text, additional discussions, and new experiments based on your valuable suggestions. Below are the key updates included in the revision:
+ Latency evaluation of performing SVD at runtime (**Appendix I**)
+ Discussion of Token-Level related works (**Appendix J**)
+ Enhanced descriptions of the experimental setup and results (**Sections 4.1, 4.5, and Table 3**)
+ Discussion of Alternative Strategies for Deriving Low-Rank Projection Matrices (**Appendix K**)
+ Theoretical Error Bound Analysis (**Appendix L**)
+ Theoretical Derivation on Compute and Memory Complexity for Computing Attention When Applying Palu (**Appendix M**)

Also, we have uploaded our code implementation in the supplementary materials.

---

### Comment · Reviewer_12bo · 2024-12-03

I would like to provide a summary of my current stance. Initially, I assigned a score of 5 to the paper, citing some unresolved issues that required further clarification. However, after reviewing the authors' detailed supplementary arguments and explanations, I feel that my concerns have been addressed, so I am inclined to raise my score to 6. This is my current position, reflecting a more favorable view of the paper, though I am still open to further considerations by the Area Chair regarding the final decision.

---

### Author Response · Authors · 2024-12-04
**Summary of Rebuttal (Many thanks to all reviewers and AC)**

Dear Reviewers and AC,

We sincerely appreciate the opportunity to clarify and address the reviewers’ feedback on our submission, Palu. Below, we summarize the major concerns raised during the review period and outline how they were addressed, respectively.

**1. Discussion on Additional Compute Requirements:**
+ Reviewers pJHR and Fsn9 expressed concerns about the additional computation required when applying Palu to RoPE-based attention.
+ In [global rebuttal](https://openreview.net/forum?id=LWMS4pk2vK&noteId=U4BX8kgVtS), we clarified the trade-off of computation for reduced memory footprint, supported by quantitative results in **Tables R3 and R6**.


**2. Discussion on Difference to Prior SVD Methods for Weight Parameter Reduction:**
+ Reviewers pjHR and 12bo sought clarification on how Palu differs from prior SVD-based methods for weight parameter reduction.
+ We provided detailed arguments in [global rebuttal](https://openreview.net/forum?id=LWMS4pk2vK&noteId=U4BX8kgVtS), highlighting Palu’s unique focus on KV-Cache compression, its fundamental change in the caching mechanism, and its specific technical contributions (e.g., manage reconstruction cost, automatic rank allocation, quantization compatibility, and efficient GPU kernel design).


**3. Discussion on Related Works**
+ Reviewer 12bo noted the need to discuss related work on token-level sparsity methods.
+ We include a new section in Appendix covering the discussion of these methods and provided experiments (**Table R1**) demonstrating the orthogonality of Palu to these line of approaches.

**4. Including an additional Mathematical Derivation on Error Bound**
+ Both reviewers 12bo and 2i3G suggested that an error bound analysis on the proposed method would further strengthen the paper.
+ We addressed this by providing the derivation and proof in the **Appendix L**, as requested.

---
**Summary of Reviewer Outcomes**

By the end of the review period, we observed that:
+ Three reviewers (12bo, Fsn9, 2i3G) lean toward acceptance, with booth 12bo and Fsn9 raising their assessments from 5 to 6 and highlighting Palu as valuable work.
+ Reviewer pJHR raised their score from 3 to 5, acknowledging the value of our work, while maintaining concerns about the trade-off between additional computation and memory footprint reduction we exploited in this work are only beneficial in the cherry-picking scenarios.

To address pJHR's remaining concern, we provided [additional summary](https://openreview.net/forum?id=LWMS4pk2vK&noteId=3QAWNiu7pe) and clarification demonstrating that our evaluations cover a large range of sequence lengths (4k~256k). We also provide benchmarking on end-to-end LLM decoding using popular LLM (e.g., Llama). We believe these results are comprehensive and genuinely show Palu can **practically accelerate LLM decoding in real-world usage**, not just for cherry-picking scenarios.

**Final Remark**

It is encouraging to see Palu recognized as a valuable contribution, with reviewers highlighting its **novelty and creativity (pjHR, 2i3G)**, **robust and general framework design (12bo, Fsn9)**, and **comprehensive evaluation (12bo)**. We sincerely express our gratitude to all reviewers and AC for their effort and time dedicated to reviewing the proposed Palu paper.

---

### Meta-Review · Area_Chair_Lgta · 2024-12-18

**Metareview:**

Dear Authors,

Thank you for your valuable contribution to the ICLR and the ML community. Your submitted paper has undergone a rigorous review process, and I have carefully read and considered the feedback provided by the reviewers.

This paper introduces a new framework for compressing the KV cache in LLMs, which reducing memory usage and improves inference. The approach employs low-rank projections, group-head low-rank decompositions, and automated rank allocation to achieve up to 50% KV cache compression with minimal accuracy loss. Overall, the paper received mostly positive response from the reviewers (6,6,6,5) scores.

Given this positive assessment, I am willing to recommend the acceptance of your paper for publication.

I would like to remind you to carefully review the reviewer feedback and the resulting discussion. While most reviews were positive, the reviewers have offered valuable suggestions that can further strengthen the quality of the paper. Please take another careful look a the 'weaknesses' section of each reviewer comment. I encourage you to use this feedback to make any necessary improvements and refinements before submitting the final version of your paper.

Once again, thank you for submitting your work to ICLR.

Best,
Area Chair

**Additional Comments On Reviewer Discussion:**

Reviewers pointed out issues in presentation and writing. They also pointed out several drawbacks of the proposed KV cache compression. In particular Reviewer pJHR questioned to extra overhead due to the introduced matrix multiplies, and also initially criticized the experimental results. After the author feedback, some of these issues were resolved and Reviewer pJHR updated their score to 5. The reviewer remains skeptical regarding the overhead created by matrix multiplies, however, all three reviewers who gave a score of 6 found the approach valuable.

---

### Decision · Program_Chairs · 2025-01-22

Accept (Poster)